# Proteolysis of fibrillin-2 microfibrils is essential for normal skeletal development

Timothy J Mead[1], Daniel R Martin[1], Lauren W Wang[1], Stuart A Cain[2], Cagri Gulec[3], Elisabeth Cahill[1], Joseph Mauch[1], Dieter Reinhardt[4], Cecilia Lo[3], Clair Baldock[2], Suneel S Apte[1]*

[1]Department of Biomedical Engineering and Musculoskeletal Research Center, Cleveland Clinic Lerner Research Institute, Cleveland, Ohio, United States; [2]Division of Cell-Matrix Biology and Regenerative Medicine, Wellcome Centre for Cell-Matrix Research, School of Biological Sciences, Faculty of Biology, Medicine and Health, The University of Manchester, Manchester Academic Health Science Centre, Manchester, United Kingdom; [3]Department of Developmental Biology, University of Pittsburgh School of Medicine, Pittsburgh, Pennsylvania, United States; [4]Faculty of Medicine and Health Sciences and Faculty of Dental Medicine and Oral Health Sciences, McGill University, Montreal, Canada

*For correspondence:
aptes@ccf.org

Competing interest: The authors declare that no competing interests exist.

**Abstract:** The embryonic extracellular matrix (ECM) undergoes transition to mature ECM as development progresses, yet few mechanisms ensuring ECM proteostasis during this period are known. Fibrillin microfibrils are macromolecular ECM complexes serving structural and regulatory roles. In mice, *Fbn1* and *Fbn2,* encoding the major microfibrillar components, are strongly expressed during embryogenesis, but fibrillin-1 is the major component observed in adult tissue microfibrils. Here, analysis of *Adamts6* and *Adamts10* mutant mouse embryos, lacking these homologous secreted metalloproteases individually and in combination, along with in vitro analysis of microfibrils, measurement of ADAMTS6-fibrillin affinities and N-terminomics discovery of ADAMTS6-cleaved sites, identifies a proteostatic mechanism contributing to postnatal fibrillin-2 reduction and fibrillin-1 dominance. The lack of ADAMTS6, alone and in combination with ADAMTS10 led to excess fibrillin-2 in perichondrium, with impaired skeletal development defined by a drastic reduction of aggrecan and cartilage link protein, impaired BMP signaling in cartilage, and increased GDF5 sequestration in fibrillin-2-rich tissue. Although ADAMTS6 cleaves fibrillin-1 and fibrillin-2 as well as fibronectin, which provides the initial scaffold for microfibril assembly, primacy of the protease-substrate relationship between ADAMTS6 and fibrillin-2 was unequivocally established by reversal of the defects in *Adamts6*[-/-] embryos by genetic reduction of *Fbn2*, but not *Fbn1*.

## Editor's evaluation

The study proves that the in vivo genetic interaction between ADAMTS6 and fibrillin2 is critical for normal endochondral bone development. In particular, the authors show that global loss of ADAMTS6 causes a severe chondrodysplasia that is significantly worsened by concomitant loss of ADAMTS10 and, conversely, almost fully prevented by haploinsufficiency for fibrillin2, a substrate of ADAMTS6. The paper expands and deepens our current understanding of proteases and their substrates in endochondral bone development.

## Introduction

In addition to proliferation and differentiation of resident cells, proper tissue and organ development, structure and function require an appropriate extracellular matrix (ECM). How ECM architecture and stoichiometry are maintained and adjusted in coordination with the dynamic tissue changes during morphogenesis and transition to the adult organism is unknown. Ontogenetically, the earliest cell collectives formed sheets and tubes with a well-established basement membrane that provided a substrate for cell migration and adhesive inputs that determined cell polarity. Expansion of ECM-encoding genes enabled formation of a complex interstitial matrix that promoted evolution of ever more complex organisms (*Huxley-Jones et al., 2007b*), but presented a challenge for remodeling of the increasingly diverse ECM repertoire within complex ECM assemblies, which appears to have been met by expansion of secreted and cell-surface protease genes (*Huxley-Jones et al., 2005*; *Huxley-Jones et al., 2007a*). For example, there are 19 ADAMTS proteases present in mammals compared to only 2 in the fruitfly and the majority cleave ECM molecules (*Dubail and Apte, 2015*; *Mead and Apte, 2018a*). However, which specific ECM molecules and supramolecular assemblies are targets of the individual proteases, the specific contexts they operate in, and relationships between individual proteases are poorly understood. The embryonic interstitial ECM is highly hydrated owing to abundant macromolecular hyaluronan (HA)-proteoglycan aggregates whereas fibrillar components, primarily collagens and elastin, dominate juvenile and adult ECM composition and provide structural resilience needed for the greater mechanical demands imposed by postnatal life. Beyond structural roles, ECM sequesters and regulates the activity of morphogens and growth factors (*Ramirez et al., 2018*; *Thomson et al., 2019*), and ECM proteolysis can generate bioactive fragments, termed matrikines (*Ricard-Blum and Vallet, 2019*).

Fibrillin microfibrils have a crucial role in tissue development and homeostasis by providing mechanical stability and limited elasticity to tissues and/or regulating growth factors of the TGFβ superfamily, including bone morphogenetic proteins (BMPs) and growth/differentiation factors (GDFs) (*Ramirez et al., 2018*; *Thomson et al., 2019*), along with a key role in elastic fiber assembly (*Shin and Yanagisawa, 2019*; *Kozel and Mecham, 2019*). Fibrillins are large, cysteine-rich glycoproteins containing many epidermal growth factor (EGF)-like repeats. Of the three fibrillin isoforms, fibrillin-2 and fibrillin-3 (in humans) are primarily expressed during embryogenesis (*Sabatier et al., 2011*; *Zhang et al., 1994*; *Zhang et al., 1995*). The gene encoding fibrillin-3 is inactivated in mice (*Corson et al., 2004*), providing a simpler scenario than in humans for investigating developmental regulation of microfibril composition and the role of proteolytic turnover therein. Among numerous gene mutations affecting the skeleton (*Yip et al., 2019*), *FBN1* and *FBN2* mutations cause Marfan syndrome and congenital contractural arachnodactyly, respectively, in humans (*Robinson et al., 2006*). Despite overlapping features such as skeletal overgrowth and poor muscular development, each disorder mainly has distinct manifestations, indicating that fibrillin isoforms either contribute specific mechanical or regulatory properties to microfibrils, have a tissue-specific function, or form distinct ECM networks. Severe cardiovascular manifestations, especially aortic root and ascending aorta aneurysms, which are potentially lethal, as well as ocular manifestations, occur frequently in Marfan syndrome, but neither is typically associated with *FBN2* mutations (*Robinson et al., 2006*). In mice, *Fbn2* deficiency affects myogenesis and distal limb patterning, indicating a key role for fibrillin-2 in BMP/GDF regulation (*Arteaga-Solis et al., 2001*; *Sengle et al., 2015*).

Fibrillin microfibrils can be homotypic or heterotypic (*Charbonneau et al., 2003*; *Marson et al., 2005*; *Lin et al., 2002*), but since each fibrillin appears to have distinct roles in vivo as well as in vitro (*Nistala et al., 2010*), intriguing questions are how the correct stoichiometry of the two fibrillins is maintained, and the impact of excess fibrillin microfibrils or altered fibrillin stoichiometry on biological systems. Here, analysis of mouse mutants of the homologous secreted metalloproteases ADAMTS6 and ADAMTS10 has identified a transcriptionally adapted system for proteostasis of fibrillin microfibrils, defined the impact of ADAMTS6 and ADAMTS10 deficiency on skeletal development, uncovered and demonstrated the importance of a specific protease-substrate relationship between these proteases and fibrillin-2 relevant to skeletal development and unearthed an impact on limb growth factor sequestration.

ADAMTS10 has established relevance to skeletal dysplasias. Recessive *ADAMTS10* mutations lead to an acromelic dysplasia, Weill-Marchesani syndrome 1 (WMS1) (*Dagoneau et al., 2004*), whereas dominant *FBN1* mutations cause a phenotypically similar disorder, WMS2 (*Faivre et al., 2003b*),

suggesting a functional relationship between ADAMTS10 and fibrillin-1 (*Hubmacher and Apte, 2015*; *Karoulias et al., 2020*). Indeed, ADAMTS10 bound fibrillin-1 directly and accelerated fibrillin-1 microfibril biogenesis in vitro, but cleaved fibrillin-1 inefficiently (*Cain et al., 2016*; *Kutz et al., 2011*). Mice with *Adamts10* inactivation or homozygous for a human WMS-causing *ADAMTS10* mutation had mildly impaired long bone growth and increased muscle mass along with reduced fibrillin-1 staining in skeletal muscle and persistence of fibrillin-2 microfibrils in skeletal muscle and the eye (*Mularczyk et al., 2018*; *Wang et al., 2019a*). ADAMTS10 undergoes inefficient processing by furin, typically a prerequisite for activation of ADAMTS proteases (*Kutz et al., 2011*). However, ADAMTS10, when constitutively activated by optimizing its furin processing site, can proteolytically process fibrillin-2 (*Wang et al., 2019a*). In contrast to ADAMTS10, ADAMTS6 is efficiently processed by furin (*Cain et al., 2016*; *Prins et al., 2018*), but its activity and functional relationship with ADAMTS10 are poorly characterized, although it is known to cleave latent TGFβ-binding protein 1, a fibrillin relative, as well as cell-surface heparan sulfate proteoglycans (*Cain et al., 2016*).

Since ADAMTS proteases cooperate in several physiological contexts where they are co-expressed (*Dubail et al., 2014*; *Enomoto et al., 2010*; *McCulloch et al., 2009*; *Mead et al., 2018b*; *Nandadasa et al., 2019*; *Nandadasa et al., 2015*) and their pairwise homology suggests the possibility of transcriptional adaptation (*El-Brolosy et al., 2019*; *Sztal and Stainier, 2020*), which has not been systematically investigated in this family, we investigated the impact of single or combined inactivation of *Adamts6* and *Adamts10* on mouse skeletal development, and defined the underlying mechanisms. The findings provide intriguing new insights into fibrillin microfibril proteostasis and mechanisms regulating skeletal development.

## Results

### Transcriptional adaptation of *Adamts6* and *Adamts10*

Previous work had suggested that knockdown of *ADAMTS10* in cultured human ARPE-19A cells increased *ADAMTS6* expression (*Cain et al., 2016*). To investigate the possibility that germline mutations of either mouse gene affected expression of the other, *Adamts6* and *Adamts10* mRNA levels were measured in limbs, heart and lungs of *Adamts6*[-/-] and *Adamts10*[-/-] mice. qRT-PCR analysis showed that *Adamts6* mRNA was reduced in limb and lung with respect to wild type levels in *Adamts6*-mutant mice [which have a missense Ser[149]Arg mutation that abrogates protein secretion, essentially comprising a knockout (*Prins et al., 2018*)], but was consistently increased in *Adamts10*[-/-] tissues (*Figure 1A*). *Adamts10* mRNA was undetectable in *Adamts10*[-/-] mice (which have an intragenic IRES-*lacZ* insertion that disrupts mRNA continuity), and increased in *Adamts6*[-/-] tissues (*Figure 1A*). Thus, each gene shows reciprocal increased expression when either is inactivated, suggestive of transcriptional adaptation. In contrast, mRNA for *Adamts17*, also linked to WMS (*Morales et al., 2009*), and its homolog *Adamts19* were unchanged in *Adamts6*- or *Adamts10*-deficient tissues (*Figure 1A*). We also asked whether ADAMTS6 and ADAMTS10 cleaved each other. Neither ADAMTS10 nor furin-optimized ADAMTS10 cleaved ADAMTS6 and ADAMTS6 did not cleave ADAMTS10 (*Figure 1—figure supplement 1*). These finding raised the possibility that *Adamts10*[-/-] and *Adamts6*[-/-] phenotypes could have been buffered by a compensating increase in the homologous mRNA and resulting protease activity. Furthermore, these genes could have cooperative functions, such as were previously identified in combined mutants of other homologous ADAMTS proteases (*Mead et al., 2018b*; *Nandadasa et al., 2019*). Both possibilities were addressed by analyzing the role of ADAMTS6 in skeletal development, characterization of the skeletal defects in *Adamts6*[-/-];*Adamts10*[-/-] mice and determination of ADAMTS6 activity in regard to fibrillins, none of which were previously undertaken. Combinations of the two mutant alleles were obtained by interbreeding, since each gene is on a distinct mouse chromosome.

### Severe skeletal malformations in *Adamts6*-deficient embryos are exacerbated by concurrent *Adamts10* inactivation

Analysis of the *Adamts6*[-/-] and *Adamts10*[-/-] crosses showed that the mutant genotypes were recovered at the expected Mendelian ratio at the end of the embryonic period (E18.5) (*Figure 1—figure supplement 2A, B*). *Adamts6*[-/-] embryos had severe reduction of crown to rump length, measured at E14.5,

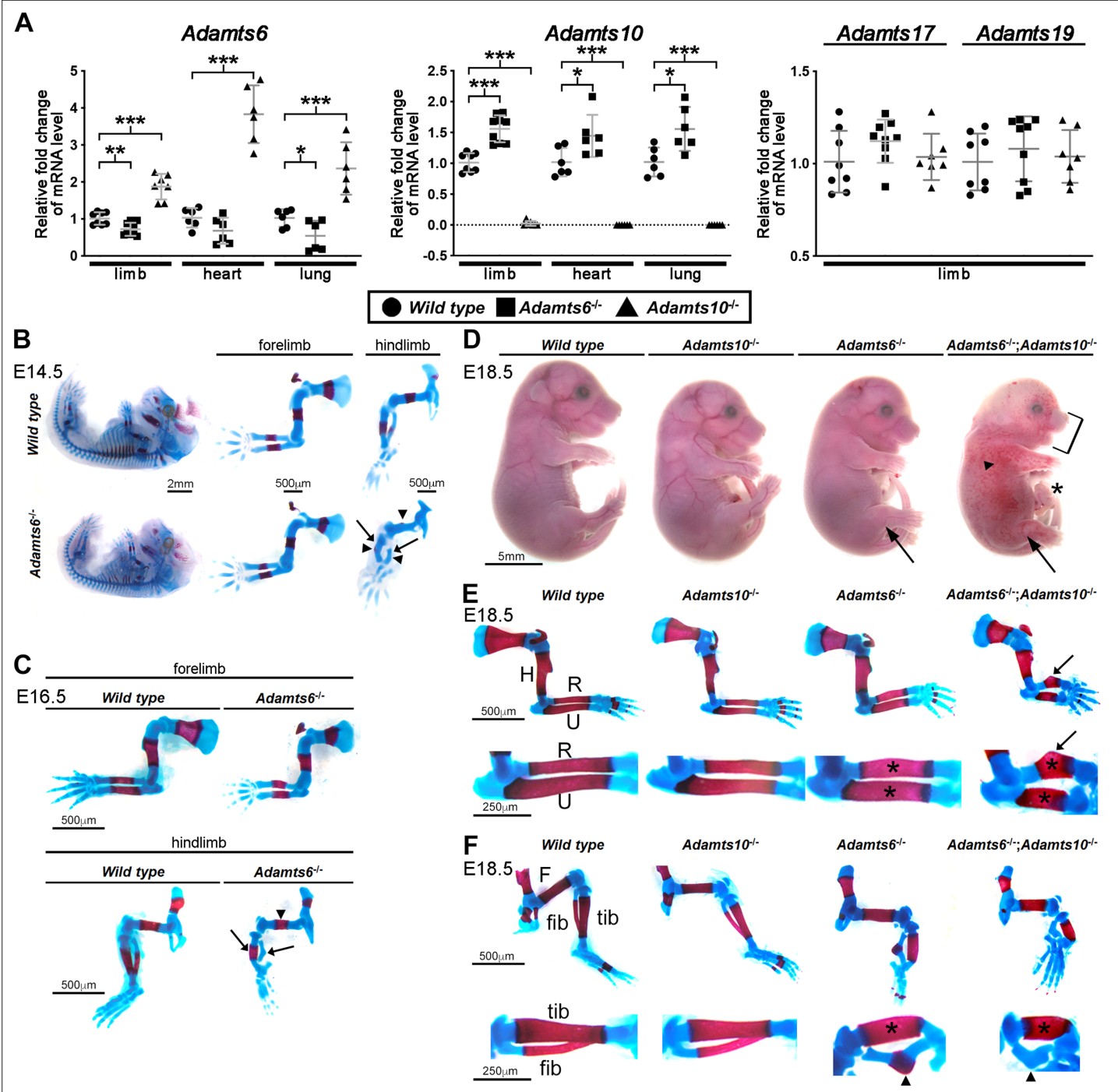

**Figure 1.** ADAMTS6 and ADAMTS10 are subject to transcriptional adaptation and cooperate in skeletal development. (**A**) qRT-PCR analysis of *Adamts6* and *Adamts10* mRNA levels in wild type, *Adamts6⁻/⁻* and *Adamts10⁻/⁻* limb, heart and lung show that *Adamts6* mRNA is elevated in *Adamts10⁻/⁻* limb, heart and lung and *Adamts10* mRNA is elevated in *Adamts6⁻/⁻* tissues. n ≥ 6. Error bars represent ± SEM. *p ≤ 0.05; **p ≤ 0.01; ***p ≤ 0.001, Student t-test. There is no change in *Adamts17* or *Adamts19* mRNA levels in the *Adamts6* or *Adamts10*-deficent limbs. (**B, C**) E14.5 (**B**) and E16.5 (**C**) alcian blue- and alizarin red-stained skeletons show severely short and under-ossified *Adamts6⁻/⁻* hindlimb long bones (arrowheads) with severely distorted tibia and fibula (arrow). *Adamts6⁻/⁻* forelimbs are not as severely affected as hindlimbs. (**D**) E18.5 *Adamts6⁻/⁻;Adamts10⁻/⁻* embryos have more severe hindlimb malformations (arrows) and shorter forelimbs and snout than *Adamts6⁻/⁻* and *Adamts10⁻/⁻* embryos as well as mandibular hypoplasia (bracket), an omphalocele (asterisk) and widespread cutaneous hemorrhage (arrowhead). (**E, F**) Alcian blue- and alizarin red-stained *Adamts6⁻/⁻;Adamts10⁻/⁻* forelimbs (**E**) and hindlimbs (**F**) show unremodeled, thicker and shorter long bones (asterisks show thicker diaphysis; H, humerus; F, femur; U, ulna; R, radius; Tib, tibia; Fib, fibula) with angulated radius (arrow) and tibia (arrowhead). Note the unossified fibula in the E18.5 *Adamts6⁻/⁻;Adamts10⁻/⁻* hindlimb.

*Figure 1 continued on next page*

*Figure 1 continued*

The online version of this article includes the following figure supplement(s) for figure 1:

**Figure supplement 1.** ADAMTS6 and ADAMTS10 do not cleave each other.

**Figure supplement 2.** Expected Mendelian ratios are observed in mouse crosses.

**Figure supplement 3.** Reduced crown-rump length and shorter limb skeletal elements in *Adamts6⁻/⁻* embryos.

**Figure supplement 4.** Reduced crown-rump length and shorter limb skeletal elements in combined *Adamts6* and *Adamts10* alleles at E18.5.

**Figure supplement 5.** Greater diaphyseal width in *Adamts6⁻/⁻* long bones.

**Figure supplement 6.** Delayed ossification and shorter femora in *Adamts6⁻/⁻* embyros.

**Figure supplement 7.** Angulation of the *Adamts6⁻/⁻* distal tibia.

**Figure supplement 8.** Severely hypoplastic fibula in *Adamts6⁻/⁻* embryos.

**Figure supplement 9.** Decreased mineralization in *Adamts6*-deficent hindlimb long bones.

**Figure supplement 10.** *Adamts6*-mutant embryos have a severely underdeveloped axial skeleton.

E16.5 and E18.5, which was also statistically significant in *Adamts10⁻/⁻* mice at E18.5 (***Figure 1—figure supplement 3A, B***, ***Figure 1—figure supplement 4A***). Whereas *Adamts6⁻/⁻* forelimbs appeared shorter, *Adamts6⁻/⁻* hindlimbs were severely short and internally rotated (***Figure 1D***, ***Figure 1—figure supplement 3A, B***, ***Figure 1—figure supplement 4B***, ***Figure 1—figure supplement 5***). Alizarin red- and alcian blue-stained skeletal preparations at embryonic day (E)14.5, when ossification is initiated, and at E16.5 and E18.5, by which time it is established, showed reduced ossification of *Adamts6⁻/⁻* hindlimb distal long bones and deformation of all hindlimb segments (***Figure 1B, C and F***), most evident in the thicker tibia and angulated fibula (***Figure 1B, C and F***, arrows). *Adamts6⁻/⁻* forelimbs demonstrated failure of diaphyseal remodeling (e.g. wider, tubular-appearing radius and ulna) and smaller ossific centers than wild type littermates after E14.5 (***Figure 1B, C and E***). Neither forelimbs nor hindlimbs showed skeletal patterning anomalies. In contrast to *Adamts6⁻/⁻* embryos, *Adamts10⁻/⁻* embryos had a morphologically normal appendicular skeleton without statistically significant shortening of individual long bones (***Figure 1—figure supplement 4***; ***Mularczyk et al., 2018***; ***Wang et al., 2019a***).

Histologic comparison of alcian blue-stained hindlimb long bone sections at daily intervals from E14.5-E18.5 showed delayed appearance of femoral and fibular primary ossific centers in *Adamts6⁻/⁻* embyros, with persistence of hypertrophic chondrocytes and delayed vascular invasion; in the tibia, the primary ossific center appeared on schedule, but was persistently smaller (***Figure 1—figure supplements 6–8***). The distal tibia and fibula showed pronounced angulation as early as E14.5, persisting through E18.5 (***Figure 1—figure supplements 7–8***). Mineral deposition, visualized by von Kossa staining at E14.5, was reduced in the perichondrium/periosteum as well as within the primary center hypertrophic zones in *Adamts6⁻/⁻* femur and tibia and undetectable in the fibula as compared to control (***Figure 1—figure supplement 9A, B***), whereas by E18.5, mineralization was seen in each primary center (***Figure 1—figure supplement 9C, D***).

The axial and craniofacial skeleton were also abnormal in E18.5 *Adamts6⁻/⁻* embryos, with shortened, tubular ribs, absence of sternal segmentation and under-ossified xiphoid processes (***Figure 1—figure supplement 10***). Additionally, their vertebral bodies were smaller in size with corresponding size reduction of all vertebral ossification centers. *Adamts6⁻/⁻* craniofacial skeletons had reduced anterior-posterior and nasal-occipital dimensions, a corresponding reduction in size of individual elements, delayed ossification of parietal bones and wider anterior fontanelles (***Figure 1—figure supplement 10***).

*Adamts6⁻/⁻;Adamts10⁻/⁻* embryos demonstrated markedly more severe external anomalies than *Adamts6⁻/⁻* mutants including subcutaneous hemorrhage, micrognathia and an omphalocele, along with more severe forelimb and hindlimb dysmorphology (***Figure 1D***). Skeletal preparations showed more severe hindlimb anomalies than in *Adamts6⁻/⁻* mutants and appearance of forelimb abnormalities similar in degree of severity to *Adamts6⁻/⁻* hindlimbs with externally evident shortening, the greatest reduction of nose-rump length among all genotypes and the shortest long bones among the generated genotypes (***Figure 1D–F***, ***Figure 1—figure supplement 4***). Fibular ossification was minimal, and the zeugopods showed pronounced tibio-fibular angulation (***Figure 1F***). The shortened and tubular ribs, vertebral bodies with smaller ossification centers, lack of xiphoid process ossification and poor

ossification of cranial bones resulting in larger fontanelles, further demonstrated more severe impairment of axial and craniofacial skeletal development than in *Adamts6*[-/-] embryos (*Figure 1—figure supplement 10*). Whereas inactivation of one *Adamts6* allele in *Adamts10*[-/-] mice did not significantly worsen the observed dysmorphology and skeletal phenotype, *Adamts10* haploinsufficiency further reduced *Adamts6*[-/-] crown to rump length and led to shorter femur, tibia, fibula, humerus, radius and ulna (*Figure 1—figure supplement 4B*).

## *Adamts6* and *Adamts10* have overlapping expression in limb skeletal development

To follow up on prior publications showing strong expression of *Adamts6* in the heart and of *Adamts10* in multiple embryonic and adult tissues (*Wang et al., 2019a*; *Prins et al., 2018*; *Somerville et al., 2004*) as well as immunohistochemical localization of ADAMTS10 in limb growth cartilage, perichondrium and muscle, we compared the spatial and temporal localization of their mRNAs during hindlimb development using RNAscope in situ hybridization. *Adamts6* and *Adamts10* exhibited overlapping expression in E13.5 resting and proliferating zone chondrocytes and perichondrium (*Figure 2A*). At E14.5, when the first primary ossific centers are formed, both genes maintained strong expression in the perichondrium (*Figure 2B*). They are expressed in the the knee joint mesenchyme, and *Adamts6* was strongly expressed in the patellar tendon (*Figure 2C*). Both genes were expressed in muscles and tendons around the hip, elbow, and ankle and in vertebral endplates (*Figure 2C and D*).

## *Adamts6*-deficient skeletal elements have abnormal growth plates with dramatically reduced cartilage proteoglycan

RGB trichrome-stained E18.5 sections revealed abnormal growth plate cartilage ECM, showing a dramatic reduction in staining with the alcian blue component, which detects sulfated proteoglycans, in *Adamts6*[-/-] and *Adamts6*[-/-];*Adamts10*[-/-] embryos (*Figure 3A*). The perichondrium and adjacent mesenchyme showed increased and diffuse staining of collagen in the *Adamts6*-deficient and *Adamts6*[-/-];*Adamts10*[-/-] femur (*Figure 3—figure supplement 1*). The reduction of alcian blue staining and diffuse collagen staining was evident in E14.5 *Adamts6*[-/-] long bone, such as the femur (*Figure 3—figure supplement 2*). *Adamts6*[-/-] and *Adamts6*[-/-];*Adamts10*[-/-] resting zone (RZ) chondrocytes were tightly packed compared to wild type with disorganized columnar zones (CZ) and hypertrophic zones (HZ), whereas *Adamts10*[-/-] growth plates resembled the wild type (*Figure 3A*). To determine the basis for reduced alcian blue staining intensity, we immunostained for aggrecan, which revealed less intense staining in *Adamts6*[-/-] and *Adamts6*[-/-];*Adamts10*[-/-] cartilage as compared to control and *Adamts10*[-/-] limbs and in E14.5 *Adamts6*[-/-] femur (*Figure 3B*, *Figure 3—figure supplement 2B, C*, *Figure 3—figure supplement 3*). Consistent with reduced aggrecan, we observed reduced staining of cartilage link protein, which stabilizes the aggregates formed by aggrecan and hyaluronan and for the key chondrogenic factor Sox9 in both E14.5 and E18.5 femoral growth plates (*Figure 3B*, *Figure 3—figure supplement 2B, C*, *Figure 3—figure supplement 3*). In contrast, the HZ marker Col X revealed comparable staining in all genotypes, and suggested increased HZ thickness in *Adamts10*[-/-], *Adamts6*[-/-] and *Adamts6*[-/-];*Adamts10*[-/-] femur as compared to wild type, potentially related to delayed ossification (*Figure 3B*, *Figure 3—figure supplement 3*). PCNA and TUNEL staining revealed no change in cell proliferation or cell death, respectively, in *Adamts6*[-/-] femoral cartilage (*Figure 3—figure supplement 4*). These findings therefore indicate a substantial disruption of the *Adamts6*-mutant chondrocyte phenotype, which was exacerbated in the *Adamts6*[-/-];*Adamts10*[-/-] mutants.

## Fibrillin-*2* and MAGP1 accumulate in *Adamts6*[-/-], *Adamts10*[-/-] and *Adamts6*[-/-];*Adamts10*[-/-] hindlimbs

Since prior work had shown that ADAMTS10 had a strong functional relationship with fibrillin microfibrils, specifically accelerating fibrillin-1 assembly and undertaking fibrillin-2 proteolysis (by the action of the small fraction of ADAMTS10 that is furin-processed) (*Kutz et al., 2011*; *Mularczyk et al., 2018*; *Wang et al., 2019a*), we investigated changes in fibrillin-1/–2 distribution and staining intensity in the mutant limbs using immunofluorescence. First, we stained for microfibril-associated glycoprotein 1 (MAGP1), which binds to both fibrillin-1 and fibrillin-2 (*Jensen et al., 2001*), to report their combined abundance and detected increased staining intensity in *Adamts10*[-/-], *Adamts6*[-/-] and *Adamts6*[-/-];*Adamts10*[-/-] femur perichondrium (*Figure 4A*, *Figure 4—figure supplement 1A*). Immunostaining

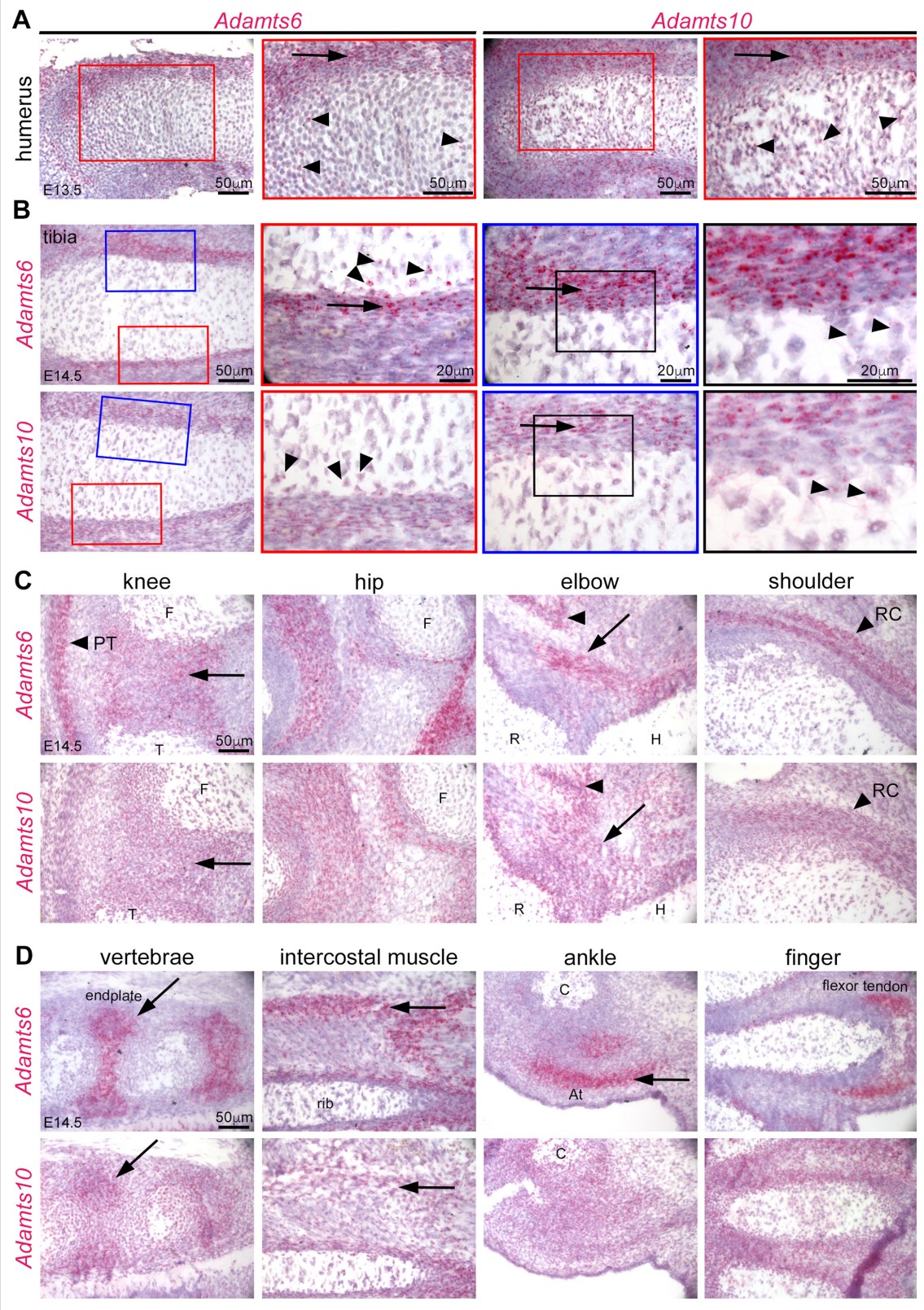

**Figure 2.** *Adamts6* and *Adamts10* mRNAs show overlapping distribution in developing mouse limbs. (**A**) RNAscope in situ hybridization shows strong expression of *Adamts6* and *Adamts10* in E13.5 perichondrium (arrow) and weaker expression in growth plate chondrocytes (arrowheads). (**B**) Strong *Adamts6* and *Adamts10* expression was seen in E14.5 tibia perichondrium, whereas only the peripheral chondrocytes expressed both genes. (**C,D**) At E14.5 *Adamts6* and *Adamts10* expression was seen, in addition to perichondrium, in knee joint mesenchyme (**C**), arrow; F, femur, T, tibia, while *Adamts6*

*Figure 2 continued on next page*

*Figure 2 continued*

was strongly expressed in the patellar tendon (**C**), arrowhead, PT. Both genes were expressed in the perichondrium, muscles and tendons around the hip, elbow tendons (arrow; H, humerus, R, radius) and muscle (arrowhead), and in the shoulder rotator cuff tendons (arrowhead, RC). (**D**) *Adamts6* and *Adamts10* were expressed in the vertebral endplates (arrows), rib intercostal muscles (arrows), Achilles tendon in the ankle (arrow; C, calcaneus), in perichondrium and tendons in the autopod.

with a monospecific fibrillin-2 antibody showed more intense perichondrial staining in *Adamts10*[-/-], *Adamts6*[-/-] and *Adamts6*[-/-];*Adamts10*[-/-] perichondrium (*Figure 4B*) but comparable fibrillin-1 staining was observed (*Figure 4C*, *Figure 4—figure supplement 1A*). *Fbn2* and *Fbn1* mRNA levels were unchanged in *Adamts6*- and *Adamts10*-deficient hindlimbs, suggesting that increased fibrillin-2 staining was not a result of increased transcription (*Figure 4—figure supplement 1B*).

## ADAMTS6 binds to fibrillin-2 microfibrils formed by cultured fibroblasts

To determine the functional relationship of ADAMTS6 to the two fibrillins, we investigated ADAMTS6 binding to homotypic fibrillin-2 or fibrillin-1 microfibrils assembled by *Fbn1*[-/-] and *Fbn2*[-/-] mouse embryo fibroblasts (MEFs), respectively. This specific approach allowed categoric examination of binding to each individual fibrillin, since when co-expressed, the fibrillins form heterotypic fibrils and thus isoform-specific binding cannot be otherwise discerned. Co-cultures of *Fbn1*[-/-] MEFs with HEK293T cells stably expressing catalytically inactive ADAMTS6 (ADAMTS6 Glu[404]Ala, referred to hereafter as ADAMTS6 EA) for 6 days illustrated specific co-localization of ADAMTS6 EA with fibrillin-2 micro-fibrils (*Figure 5A*, top row). When *Fbn1*[-/-] MEFs were co-cultured with HEK293F cells expressing active ADAMTS6 (*Figure 5A*, 2nd row), no fibrillin-2 staining was seen, suggestive of proteolytic degradation of fibrillin-2 microfibrils or interference with their assembly. Similarly, 4-day-old cultures of *Fbn2*[-/-] mouse skin fibroblasts (MSF) revealed co-localization of ADAMTS6 EA with fibrillin-1 and demonstrated absence of fibrillin-1 microfibrils in the presence of ADAMTS6 (*Figure 5A*, center panels). Since a fibronectin fibrillar matrix is formed soon after plating of cells and acts as a template for fibrillin microfibril assembly (*Beene et al., 2013*; *Sabatier et al., 2009*), we similarly analyzed fibronectin staining, observing that ADAMTS6 EA co-localized with fibronectin and that fibronectin fibrils were absent in the presence of ADAMTS6, suggesting also that over-expressed ADAMTS6 may cleave fibronectin (*Figure 5A*, bottom panels). In agreement, MEFs isolated from *Adamts6*[-/-] embryos showed greater fibrillin-2, fibrillin-1, and fibronectin microfibril staining intensity than wild type MEFs (*Figure 4—figure supplement 1A*, *Figure 5—figure supplement 1*). Together, these findings suggested that ADAMTS6 binds to and proteolytically degrades fibronectin and fibrillin microfibrils or alternatively, unassembled fibrillin monomers prior to their assembly.

Since fibrillin microfibrils contain additional components beside fibrillins (*Cain et al., 2006*; *De Maria et al., 2017*; *Fujikawa et al., 2017*; *Mecham and Gibson, 2015*), which could bind to ADAMTS6, we next asked whether purified ADAMTS6 constructs bound directly to purified fibrillin-2 and fibrillin-1 in a sensitive binary interaction assay. Since ADAMTS proteases typically bind to their substrates via interactions of their C-terminal ancillary domains, and because of the difficulty in purifying full-length ADAMTS6, we used C-terminal ADAMTS6 constructs for these studies. Biacore analysis showed that C-terminal ADAMTS6 constructs (ADAMTS6-Ct, ADAMTS6-S4TSR and ADAMTS6-4TSR) bound the C-terminal half of fibrillin-2 (fibrillin-2-Ct) (*Figure 6A–C*, *Table 1*). Since all 4TSR-array-containing constructs bound to fibrillin-2-Ct we concluded that the fibrillin-2 binding region of ADAMTS6 was located in the C-terminal TSR array. In reciprocal Biacore analysis using ADAMTS6-Ct as the immobilized ligand, its binding to both the N- and C-terminal halves of fibrillin-2 was observed (*Figure 6D*, *Table 2*). Comparable $K_D$ values of 43 nM for fibrillin-2-Nt and 80 nM for fibrillin-2-Ct suggested two or more binding sites in each half of fibrillin-2 with similar affinities for ADAMTS6. Alternatively, the ADAMTS6 binding site on fibrillin-2 may lie in the overlapping region of the two fragments between cbEGF22 and cbEGF24. Consistent with prior work showing ADAMTS6 binding to fibrillin-1 (*Cain et al., 2016*), ADAMTS6-Ct bound to fibrillin-1 N- and C-terminal halves (*Figure 6—figure supplement 1A, B*), although stronger binding was observed to the N-terminal half (*Figure 6—figure supplement 1C*). In summary, these high-affinity binary interactions suggested that ADAMTS6 binds directly to both fibrillin-1 and fibrillin-2 to cleave them.

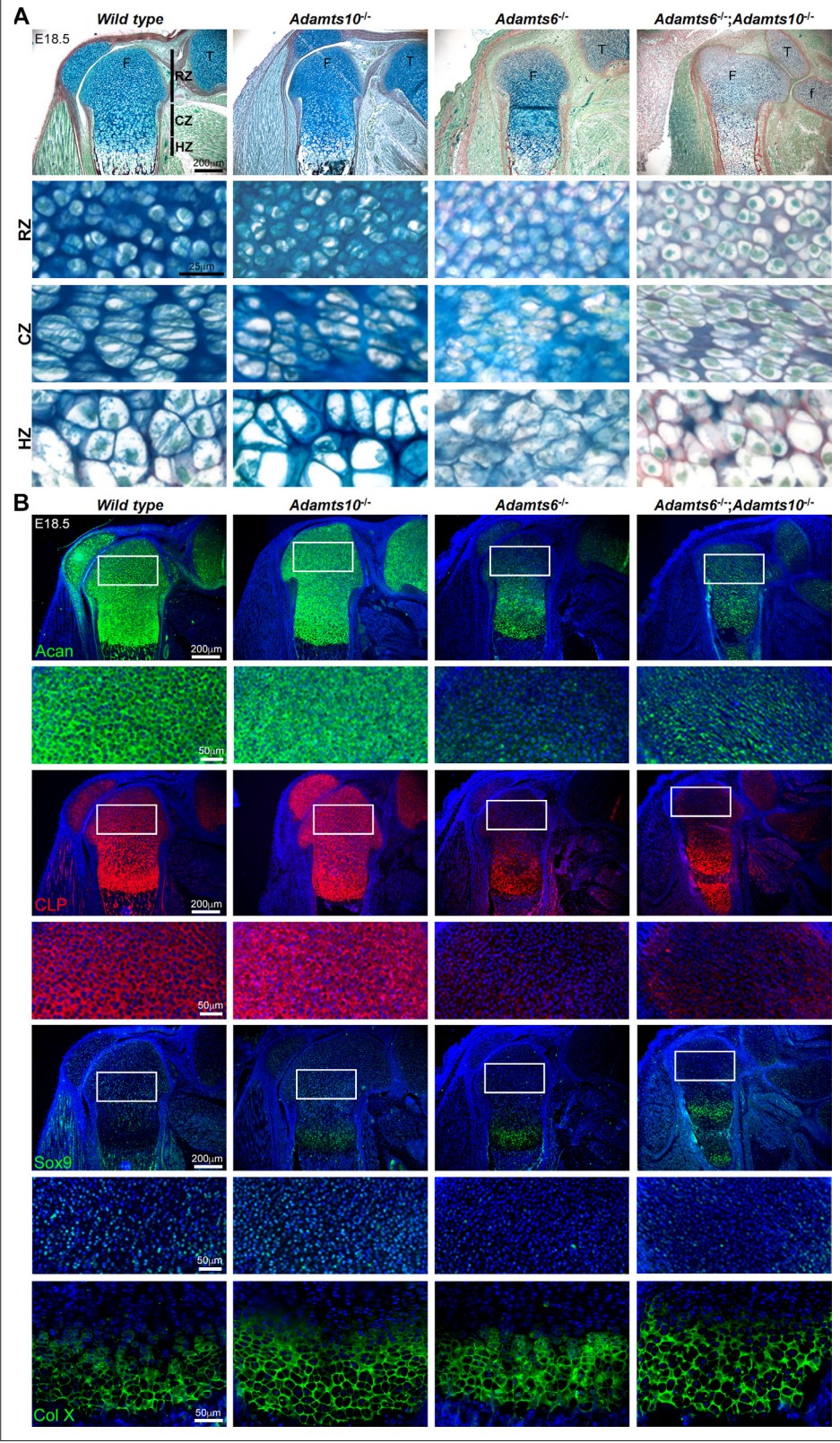

**Figure 3.** *Adamts6*[-/-] and *Adamts6*[-/-];*Adamts10*[-/-] hindlimbs have disorganized chondrocytes and reduced cartilage proteoglycan. (**A**) RGB trichrome-stained E18.5 knees show disorganized chondrocytes and reduced alcian blue staining in *Adamts6*[-/-] and *Adamts6*[-/-];*Adamts10*[-/-] bones. (F), femur; T, tibia; f, fibula; RZ, resting zone; CZ, columnar zone; HZ, hypertrophic zone. (**B**) Severely reduced aggrecan (Acan), cartilage link protein (CLP) and

*Figure 3 continued on next page*

*Figure 3 continued*

Sox9 in *Adamts6⁻/⁻* and *Adamts6⁻/⁻;Adamts10⁻/⁻* cartilage and wider collagen X (Col X)-stained region indicating an expanded HZ.

The online version of this article includes the following figure supplement(s) for figure 3:

**Figure supplement 1.** Increased collagen staining in *Adamts6*-deficient hindlimbs.

**Figure supplement 2.** *Adamts6*-deficient femora have reduced cartilage proteoglycan staining.

**Figure supplement 3.** Late embryonic *Adamts6*-mutant femora have reduced cartilage proteoglycan.

**Figure supplement 4.** No change in proliferation or cell death in *Adamts6⁻/⁻* cartilage.

## N-terminomics identification and orthogonal validation of an ADAMTS6 cleavage site in fibrillin-2

Direct binding of ADAMTS6 constructs to recombinant fibrillin-2, ADAMTS6 EA co-localization with fibrillin-2 microfibrils, loss of fibrillin-2 microfibrils in vitro in the presence of wild-type ADAMTS6, and increased fibrillin-2 staining in *Adamts6*-deficient hindlimbs suggested that fibrillin-2 was likely to be an ADAMTS6 substrate. To define the cleavage site(s), HEK293F cells stably expressing the N- or C-terminal halves of fibrillin-2 were transfected with either ADAMTS6 or ADAMTS6 EA and the serum-free conditioned medium was collected for Terminal Amine Isotopic Labeling of Substrates (TAILS) (*Figure 7A*), an N-terminomics strategy for identifying protease substrates and cleavage sites (*Kleifeld et al., 2010*; *Grant and Li, 2016*). This strategy was previously applied for identification of substrates of other ADAMTS proteases (*Bekhouche et al., 2016*). Proteins were labeled with stable isotopes of formaldehyde (natural ($CH_2O$)/light isotope applied to the ADAMTS6-containing medium or isotopically heavy ($^{13}CD_2O$), applied to the ADAMTS6 EA-containing medium). The ensuing reductive dimethylation labels and blocks free protein N-termini as well as lysine side-chains. Labeled proteins from each pair of ADAMTS6/ADAMTS6 EA digests were combined, digested with trypsin to obtain peptides and mixed with hyperbranched polyglycerol aldehyde polymer which binds to free N-termini of tryptic peptides to exclude them (*Figure 7A*), enriching the peptides having blocked and labeled N-termini for liquid chromatography tandem mass spectrometry (LC-MS/MS). Following LC-MS/MS, we undertook a targeted search for fibrillin-2 peptides with isotopically labeled N-termini having a statistically significant light:heavy ratio. This analysis revealed a putative cleavage site at the $Gly^{2158}$-$His^{2159}$ peptide bond in a linker between TGFβ binding-like domain 6 and calcium-binding epidermal growth factor (cbEGF) repeat 32 in the C-terminal half of fibrillin-2 (*Figure 7B–F*). The sequence of the indicator peptide, which was semi-tryptic, and therefore unlikely to result from trypsin digestion during sample preparation and the presence of an N-terminal label was determined with high confidence by the MS$^2$ spectrum (*Figure 7C*). Quantification of the relative ion abundance of isotoptically labeled peptides showed a considerable excess of this peptide in the presence of active ADAMTS6 (*Figure 7D–E*). Western blot of the medium from these experiments for orthogonal validation of the cleavage showed that ADAMTS6 cleaved the C-terminal half of fibrillin-2 (*Figure 7G*). The cleavage products of 100 kDa and 75 kDa matched the predicted cleavage fragments in size and their sum matched the expected mass of the FBN2-Ct construct (175 kDa). Importantly, the $Gly^{2158}$-$His^{2159}$ cleavage site is located between two folded, disulfide-bonded domains, predicting separation of the resulting fragments (as opposed to cleavage sites within disulfide-bonded fibrillin-2 domains that would remain linked). The cleavage can potentially interfere with microfibril assembly, which is reliant on multimerization via C-terminal interactions downstream of the cleavage site (*Hubmacher et al., 2008*).

## ADAMTS6 cleaves fibrillin-1 and fibronectin

Because ADAMTS6 also bound directly to fibrillin-1, TAILS was applied in a similar approach as above to determine if ADAMTS6 cleaved fibrillin-1 (*Figure 7—figure supplement 1A*). A targeted search for fibrillin-1 peptides with labeled N-termini revealed a putative cleavage site at the $R^{516}$-$A^{517}$ peptide bond in a linker between EGF4 and cbEGF3 in the N-terminal half (*Figure 7—figure supplement 1B*). The peptide sequence and presence of an N-terminal label was determined with high confidence by the MS$^2$ spectrum and quantification of the ion abundance showed a considerable excess of this peptide in the presence of ADAMTS6 (*Figure 7—figure supplement 1C-E*).

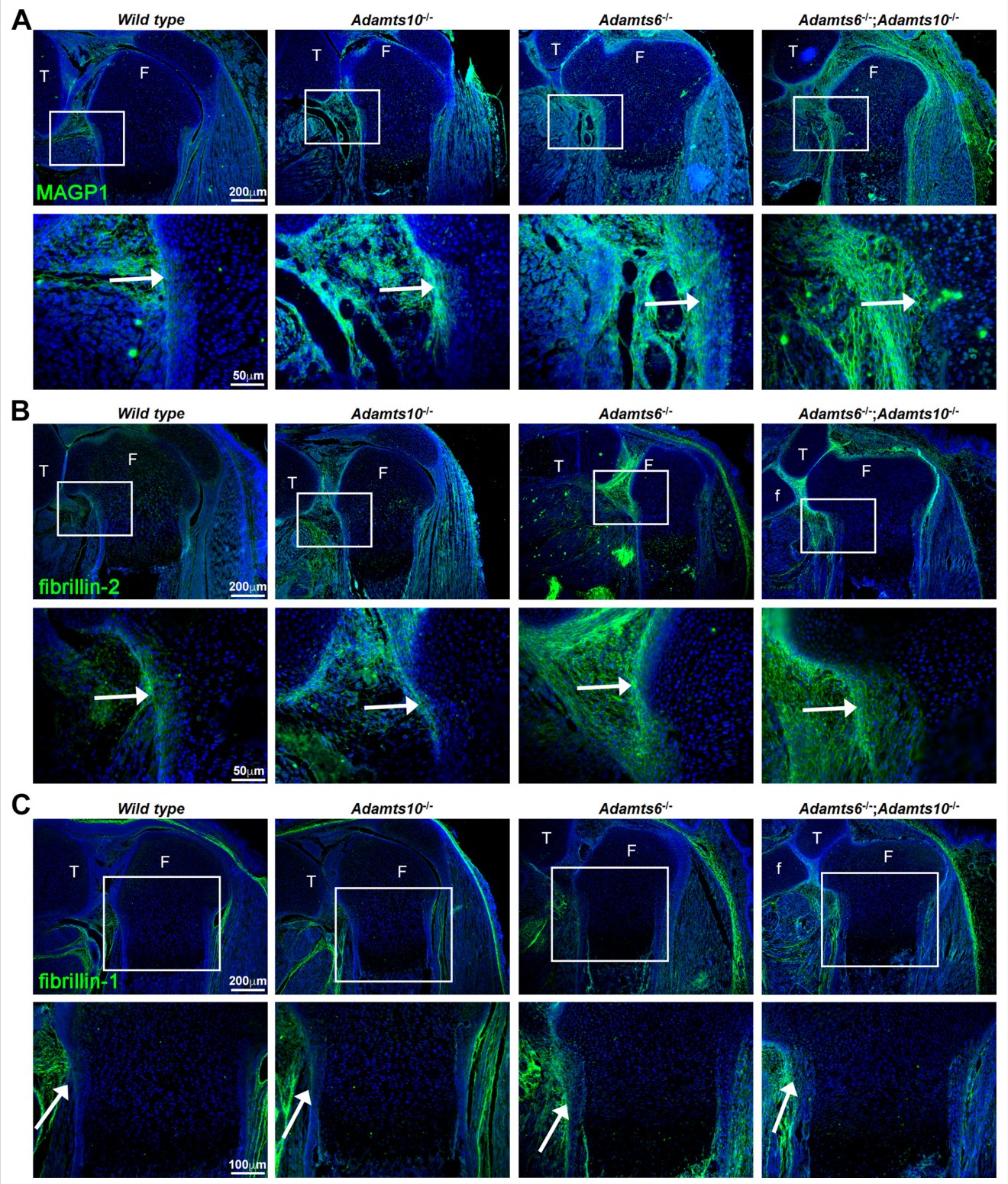

**Figure 4.** Increased MAGP1 and fibrillin-2 staining, but not fibrillin-1 staining in *Adamts6*[-/-] and *Adamts6*[-/-];*Adamts10*[-/-] perichondrium. (**A–B**) Increased staining intensity (green) of MAGP1 (**A**) and fibrillin-2 (**B**) in E18.5 *Adamts6*- and *Adamts10*-deficient knee joints. (**C**) No consistent change in fibrillin-1 staining (green) was seen between the various genotypes. n = 3. Sections are counterstained with DAPI (blue). The inset boxes are enlarged in corresponding lower sections. White arrows indicate the perichondrium. F, femur; T, tibia; f, fibula.

The online version of this article includes the following figure supplement(s) for figure 4:

**Figure supplement 1.** Differential microfibril staining in *Adamts6*-deficient hindlimbs.

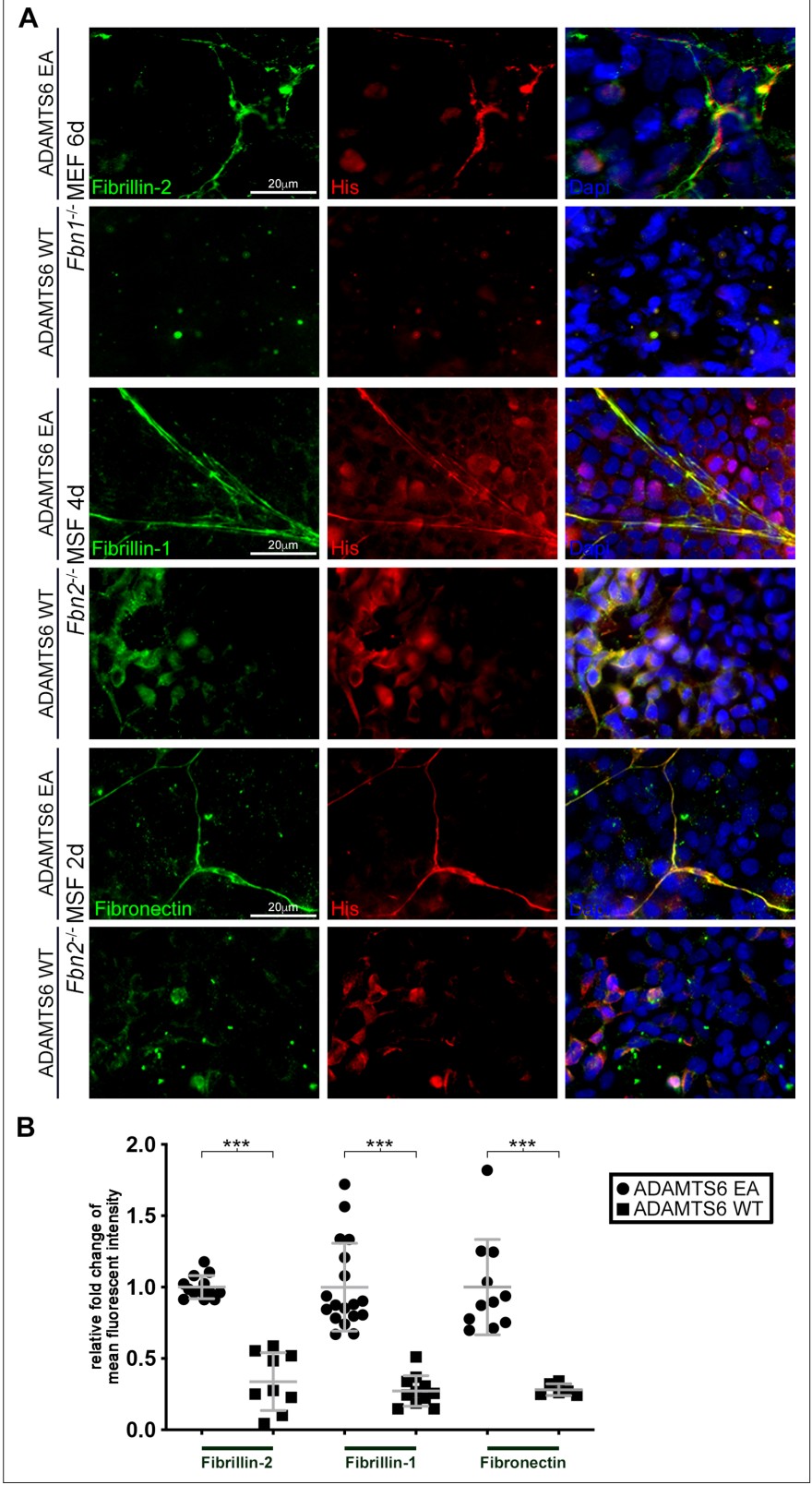

**Figure 5.** Loss of fibrillin-2, fibrillin-1 and fibronectin microfibrils in the presence of active ADAMTS6. (**A**) *Fbn1⁻/⁻* mouse embryo fibroblasts (MEFs), which can produce fibrillin-2 but not fibrillin-1, were cocultured with human embryonic kidney (HEK) cells overexpressing ADAMTS6 or ADAMTS6 EA (inactive ADAMTS6). Fibrillin-2 (green) co-localized with histidine (His)-tagged ADAMTS6 EA (red), but no microfibrils were seen in the presence of

*Figure 5 continued on next page*

*Figure 5 continued*

ADAMTS6 (red). Similarly, *Fbn2*-/- mouse skin fibroblasts (MSFs), which can produce fibrillin-1 but not fibrillin-2, were co-cultured with HEK cells overexpressing ADAMTS6 or ADAMTS6 EA. Fibrillin-1 microfibrils (green) co-localized with His-tagged ADAMTS6 EA (red), but were absent in the presence of active ADAMTS6 (red) after 4 days culture. Fibronectin also co-localized with ADAMTS6 EA, and fibronectin fibrils were absent in the presence of ADAMTS6. Nuclei are stained with DAPI (blue). (**B**) Quantification of mean intensity of fluorescence. n ≥ 6. *p ≤ 0.05; **p ≤ 0.01; ***p ≤ 0.001.

The online version of this article includes the following figure supplement(s) for figure 5:

**Figure supplement 1.** Increased fibrillin-1, fibrillin-2 and fibronectin microfibril staining in fibroblasts lacking ADAMTS6.

To determine whether ADAMTS6 cleaved fibronectin, ADAMTS6 or ADAMTS6 EA-expressing stable cell lines were plated on human fibronectin-coated plates and the serum-free medium from these cultures was collected 24 hr later. By comparing the ratio of light:heavy dimethyl-labeled peptides, TAILS revealed several potential cleavage sites in the medium from the ADAMTS6 sample (*Figure 7—figure supplement 2A*) (*Table 3*). As an example, annotated spectra of two of the peptides and their extracted ion chromatograms illustrate the lack of the corresponding peaks in medium of the ADAMTS6 EA samples, indicating fibronectin cleavage solely in the presence of catalytically active ADAMTS6 (*Figure 7—figure supplement 2B-E*).

## Fibrillin-2 reduction, but not fibrillin-1 reduction specifically reverses morphogenetic and molecular defects in *Adamts6*-/- mice

Identification of fibrillin-1, fibrillin-2 and fibronectin as ADAMTS6 substrates, yet observation of only fibrillin-2 accumulation in the hindlimbs suggested that while all three ECM components could be cleaved by ADAMTS6, fibrillin-2 could either be a preferred substrate, or not as extensively cleaved by other proteases active during limb development, which may compensate for the absence of ADAMTS6. We therefore undertook a genetic approach to determine the importance of fibrillin-2 and fibrillin-1 with regards to ADAMTS6 activity. To test whether fibrillin-2 had a significant role in the observed anomalies, fibrillin-2 was reduced genetically in *Adamts6*-/- mice. *Fbn2* haploinsufficiency dramatically restored the external craniofacial and hindlimb morphology of *Adamts6* mutants as well as the maturity, length and shape of skeletal components of the hindlimb and axial skeleton (*Figure 8A and B*, *Figure 8—figure supplement 1A, B*, *Figure 8—figure supplement 2A*). Specifically, alizarin red and alcian blue-stained skeletal preparations demonstrated reversal of reduced crown-rump length, long bone shortening and improved ossification of hindlimb long bones and ribs, an appropriately segmented sternum with a normal xiphoid process and normal fontanelle closure (*Figure 8A and B*, *Figure 8—figure supplement 1*, *Figure 8—figure supplement 2A*). In addition, cleft secondary palate which occurred with a high incidence in *Adamts6*-/- embryos, was not observed in *Adamts6*-/-;*Fbn2*+/- embryos (*Figure 8—figure supplement 2B*). *Adamts6*-/-;*Fbn2*+/- hindlimb fibrillin-2 staining showed reduced intensity comparable to wild type hindlimb (*Figure 8C*, *Figure 8—figure supplement 3*). Aggrecan, cartilage link protein and Sox9 staining intensity in *Adamts6*-/-;*Fbn2*+/- femur growth plates were restored to comparable levels as in wild type limbs, indicating restoration of the appropriate ECM of cartilage (*Figure 8D*, *Figure 8—figure supplement 3*). *Fbn2*+/- embryos had normal hindlimb development and growth plate zone organization as wild type controls (*Figure 8—figure supplement 3*, *Figure 8—figure supplement 4C-E*). In contrast to *Adamts6*-/-;*Fbn2*+/- embryos, *Adamts6*-/- embryos with ~80% reduction in fibrillin-1 levels (*Adamts6*-/-;*Fbn1*mgR/mgR embryos) had no amelioration of limb defects (*Figure 8—figure supplement 5A, B*). Taken together, these genetic analyses using well-characterized *Fbn1* and *Fbn2* mutants, suggested a specific role for ADAMTS6 in modulating fibrillin-2 abundance during limb microfibril proteostasis.

## *Adamts6*-/- hindlimbs and cartilage have abnormal GDF5 sequestration and reduced BMP signaling respectively

Since fibrillins are implicated in signaling of TGFβ and BMP/GDF signaling via sequestration and activation of these growth factors (*Sengle et al., 2015*; *Nistala et al., 2010*; *Furlan et al., 2021*; *Ramirez and Rifkin, 2009*; *Wohl et al., 2016*), we investigated the impact of ADAMTS6 mutation on TGFβ and

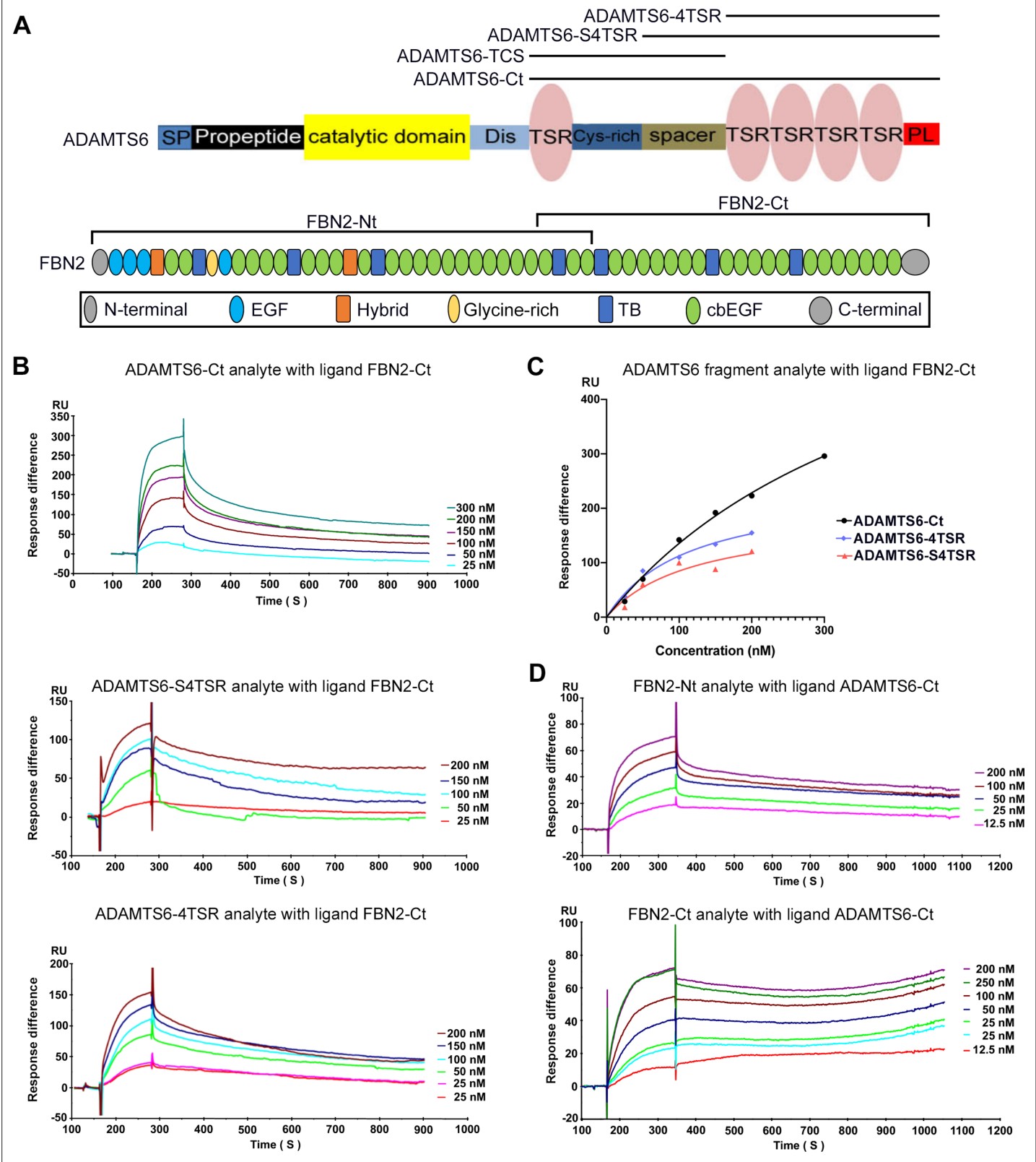

**Figure 6.** ADAMTS6 binds directly to fibrillin-2. (**A**) Cartoons of the domain structures of ADAMTS6 and fibrillin-2 and the recombinant constructs used in the present work (indicated by black lines). (**B–C**) Biacore analysis shows dose-dependent binding curves for the ADAMTS6-C-terminal constructs ADAMTS6-Ct, ADAMTS6-4TSR and ADAMTS6-S4TSR against immobilized FBN2-Ct (**B**), and comparative binding characteristics of the constructs (**C**),

*Figure 6 continued on next page*

*Figure 6 continued*

other than ADAMTS6-TCS, which did not bind. (**D**) Reciprocal Biacore analysis using immobilized ADAMTS6-Ct shows that fibrillin-2-Nt and fibrillin-2-Ct used as the analyte each bound strongly to ADAMTS6-Ct.

The online version of this article includes the following figure supplement(s) for figure 6:

**Figure supplement 1.** ADAMTS6 binds directly to fibrillin-1.

GDF/BMP signaling. *Adamts6^{-/-}* hindlimb long bones had reduced pSmad1/5/8 staining compared to wild type, which was restored to wild type levels in *Adamts6^{-/-};Fbn2^{+/-}* hindlimbs (*Figure 9A, B*). Similarily, pSMAD5 was reduced in *Adamts6^{-/-}* hindlimb extracts and restored to normal levels in *Adamts6^{-/-};Fbn2^{+/-}* hindlimb extracts (*Figure 9C, D*). In contrast, no change in TGFβ signaling, as measured by pSMAD2 immunostaining and western blot, occurred in *Adamts6^{-/-}* hindlimbs (*Figure 9E–H*).

Several BMPs and GDFs bind fibrillin-2 directly through their propeptides (*Furlan et al., 2021*; *Wohl et al., 2016*; *Gregory et al., 2005*; *Sengle et al., 2008*; *Sengle et al., 2011*). We therefore considered the possibility that increased fibrillin-2 in *Adamts6*-deficient hindlimbs resulted in sequestration of BMPs/GDFs. Antibodies applicable for immunostaining of mouse tissues are available for very few of these; accordingly, we co-immunostained sections with a fibrillin-2 antibody and a GDF5 antibody. We consistently observed stronger co-localized fibrillin-2 and GDF5 signal in *Adamts6* mutant knee joint mesenchyme, as well as in the perichondrium and interjoint regions in distal limbs, which are at less mature stages of development and have abundant undifferentiated limb mesenchyme (illustrated in knee joint interzone in *Figure 10A–C*, *Figure 10—figure supplement 1A*). Thus, increased fibrillin-2 was associated with increased GDF5 sequestration in *Adamts6*-deficient limbs.

## Discussion

Regulated ECM proteostasis is required for ensuring proper adult tissue function (mitigating the risk of organ fibrosis, for example), but presumably also occurs during transition from the early to late embryo ECM, and may be a major determinant of adaptation of embryonic tissues to the dramatically different postnatal mechanical and regulatory landscapes. Because of the diversity of molecules and networks that comprise the ECM and existence of numerous secreted proteases, we hypothesized that selective proteolysis by specific proteases having preferred ECM substrates, may have a crucial role in ECM proteostasis in specific contexts of embryonic development. One example of a specific ECM proteostatic phenomenon mediated by ADAMTS proteases that influences embryonic cell behavior and is strongly supported by mouse genetic models is their role in versican turnover during interdigital web regression (*Dubail et al., 2014*; *McCulloch et al., 2009*; *Nandadasa et al., 2021*). Other than this example, little is known about the fate of macromolecular ECM complexes, especially those formed by homologous proteins with different stoichiometry in the embryo versus adult, and their proteostatic mechanisms.

Here, using a combination of genetic, biochemical, cell culture and proteomics approaches, we have identified ADAMTS6 and ADAMTS10 as major participants in fine-tuning of fibrillin microfibrils during limb development. Although prior work linked ADAMTS10 to genetic conditions that also resulted from *FBN1* mutations, the existence of a closely related homolog, ADAMTS6, raised questions about the role of transcriptional adaptation in this protease pair, their overlapping functions on an organ, tissue and molecular scale, the identity of their specific targets and relevant function in regard to fibrillin-1 and fibrillin-2. Resolution of these questions provides a conceptual precedent and strategic framework for elucidation of embryonic

**Table 1.** Kinetic data for ADAMTS6 construct binding to FBN2-Ct.

Ligand: FBN2-Ct

| Analyte | $K_D$ (nM) | $R^2$ | $B_{max}$ (RU) |
|---|---|---|---|
| ADAMTS6-Ct | 436 | 0.9951 | 727 |
| ADAMTS6-4TSR | 122 | 0.8864 | 188 |
| ADAMTS6-S4TSR | 114 | 0.9819 | 241 |

**Table 2.** Kinetic data for binding of FBN2 constructs to ADAMTS6-Ct.

Ligand: ADAMTS6-Ct

| Analyte | $K_D$ (nM) | $R^2$ | $B_{max}$ (RU) |
|---|---|---|---|
| FBN2-Nt | 43 | 0.9972 | 85 |
| FBN2-Ct | 80 | 0.9933 | 100 |

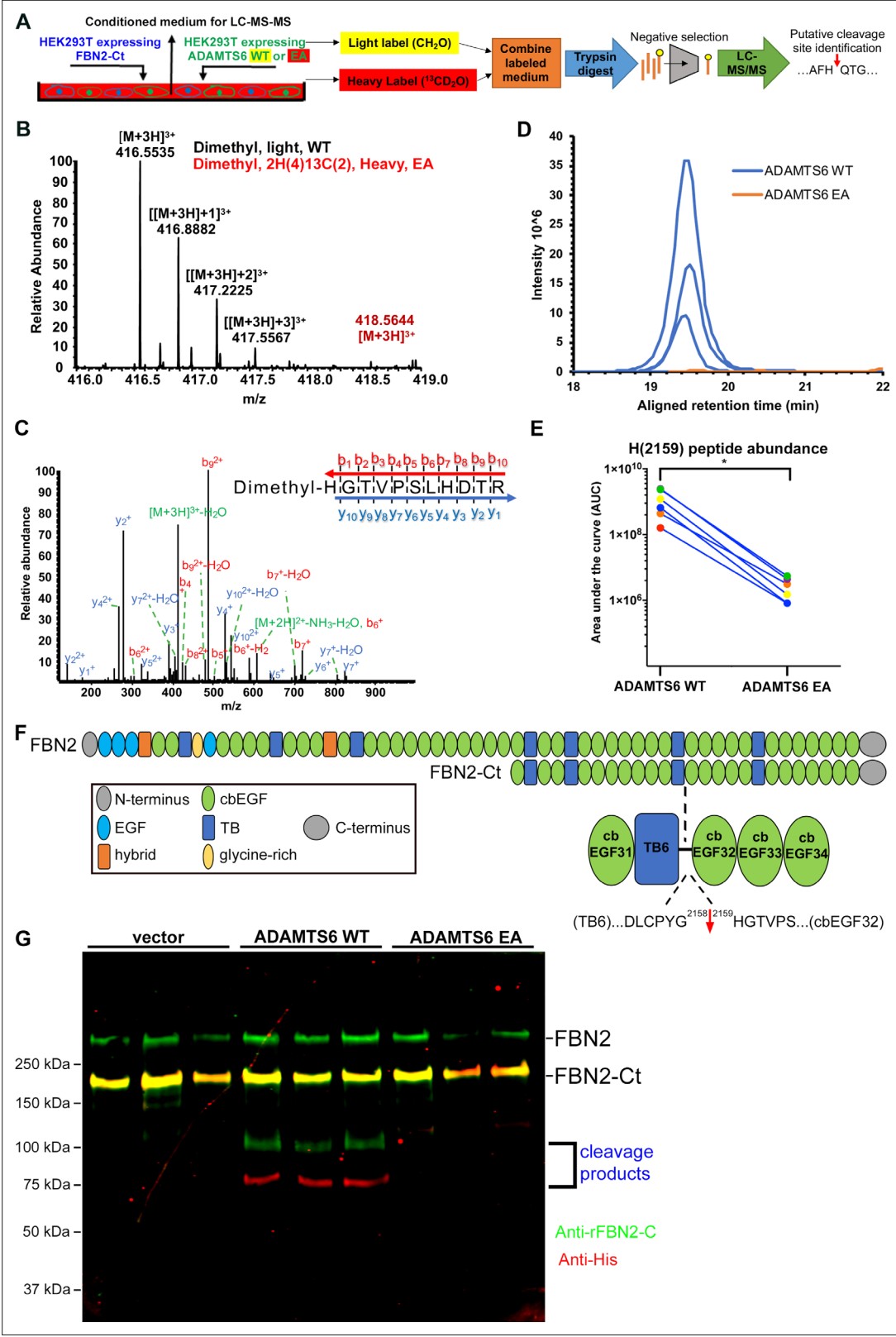

**Figure 7.** Fibrillin-2 cleavage by ADAMTS6 and identification of the cleavage site using N-terminomics. (**A**) Schematic of the experimental approach. Proteins from conditioned medium of co-cultures of HEK293F cells stably expressing FBN2-Ct and HEK293F cells expressing either ADAMTS6 WT or ADAMTS6 EA (inactive control) were labeled by reductive dimethylation using stable formaldehyde isotopes and analyzed by LC-MS/MS in the TAILS workflow described in detail in the Methods section. (**B**) MS1 chromatogram of the parent ions derived from fibrillin-2 fragments potentially resulting

*Figure 7 continued on next page*

*Figure 7 continued*

from cleavage by ADAMTS6 (black) and the same peptides sought in ADAMTS6 EA-containing medium (red) that were used for measuring abundance. (**C**) Annotated MS$^2$ spectrum of the light (ADAMTS6-generated) dimethyl peptide, showing b-(N-terminus preserved) and y-type (C-terminus preserved) ions generated by amide bond cleavage during collisional-induced dissociation that were used to derive the fibrillin-2 peptide sequence indicated at the top right. (**D**) Retention time-aligned extracted ion chromatograms (EICs) comparing abundance of the light dimethyl-labeled HGTVPSLHDTR peptide (blue) in ADAMTS6 medium and isotopically heavy dimethyl-labeled peptide (orange) in ADAMTS6 EA medium from three replicate TAILS experiments. (**E**) The area under the EIC curves was quantified and comparison of ion abundance is shown in a dumb-bell plot (from 3 TAILS and 3 pre-TAILS samples). Significance was determined using a two-tailed, paired Student t-test, * indicates p-value < .05. (**F**) Domain structure of fibrillin-2 and the C-terminal construct FBN2-Ct showing the location of the cleaved peptide bond Gly$^{2158}$-His$^{2159}$ in the linker between TB6 and cbEGF32. (**G**) Orthogonal validation of fibrillin-2 cleavage by ADAMTS6 using western blot analysis of the conditioned medium from A, shows distinct molecular species (100 kDa and 75 kDa) reactive with anti-fibrillin-2-Ct antibody (green, N-terminal fragment of fibrillin-2-Ct) and C-terminal anti-His$_6$ antibody (red, C-terminal fragment of fibrillin-2-Ct), respectively, obtained in the presence of ADAMTS6 WT, but not ADAMTS6 EA, indicative of fibrillin-2-Ct cleavage. The green band of ~350 kDa is endogenous fibrillin-2 produced by HEK293F cells. The yellow band at ~175 kDa indicates overlapping anti-His$_6$ and anti-fibrillin-2-Ct antibody staining of the fibrillin-2-Ct construct. Cells transfected with empty vector were also used to obtain control medium.

The online version of this article includes the following figure supplement(s) for figure 7:

**Figure supplement 1.** ADAMTS6 cleaves fibrillin-1.

**Figure supplement 2.** ADAMTS6 cleaves fibronectin at multiple sites.

ECM proteostatic mechanisms, about which little is known relative to cellular homeostatic mechanisms in the embryo.

Prior work showed that transcriptional adaptation is operational in genes encoding paired homologous proteases ADAMTS7 and ADAMTS12, and is potentially a broad phenomenon in this family, whose mammalian members have arisen largely from gene duplication (*Huxley-Jones et al., 2005*). Although the present study illustrates application of the phenomenon to the homologous pair of *Adamts6* and *Adamts10*, the two proteases are clearly distinct in their biological functions as outlined in prior and present work. While the combined knockouts clearly had a more severe phenotype (indicating cooperative function), it is presently not possible to demarcate the relative contributions of transcriptional buffering to mitigation of the individual phenotypes.

In the context of mechanisms that would ensure fibrillin-1 predominance postnatally, prior RNA in situ hybridization analysis had demonstrated dramatic reduction of *Fbn2* mRNA expression after birth, with *Fbn1* expression continuing (*Zhang et al., 1995*) and corresponding to this, little fibrillin-2 is detected immunohistochemically or by proteomics in adult mouse tissues (*De Maria et al., 2017*; *Hubmacher et al., 2008*; *Cain et al., 2005*; *Dallas et al., 2000*; *Kettle et al., 1999*; *Kinsey et al.,*

**Table 3.** Putative sites of fibronectin cleavage by ADAMTS6 determined using TAILS.
The cleavage site is indicated by the period in column 1, with flanking amino acids numbered.

| Annotated sequence | Number of PSMs | Wild type abundance | EA abundance |
|---|---|---|---|
| [A$^{292}$].V$^{293}$YQPQPHPQPPPYGHCVTDSGVVYSVGMQWLK.[T] | 2 | 1.38E + 06 | 0 |
| [Y$^{294}$].Q$^{295}$PQPHPQPPPYGHCVTDSGVVYSVGMQWLK.[T] | 1 | 6.99E + 05 | 0 |
| [P$^{462}$].M$^{463}$AAHEEICTTNEGVMYR.[I] | 2 | 1.22E + 06 | 0 |
| [A$^{464}$].A$^{465}$HEEICTTNEGVMYR.[I] | 1 | 4.71E + 05 | 0 |
| [G$^{567}$].T$^{568}$FYQIGDSWEK.[Y] | 1 | 1.18E + 06 | 0 |
| [F$^{750}$].R$^{751}$VEYELSEEGDEPQYLDLPSTATSVNIPDLLPGRK.[Y] | 1 | 3.33E + 05 | 0 |
| [K].YIVNVYQISEDGEQS$^{800}$.[L$^{801}$] | 1 | 8.57E + 05 | 0 |
| [Y$^{1248}$].T$^{1249}$VKDDKESVPISDTIIPAVPPPTDLR.[F] | 1 | 1.21E + 06 | 0 |
| [Y$^{1613}$].A$^{1614}$QNPSGESQPLVQTAVTTIPAPTDLK.[F]$^†$ | 1 | 6.32E + 05 | 0 |
| [P$^{2021}$].F$^{2022}$VTHPGYDTGNGIQLPGTSGQQPSVGQQMIFEEHGFRR.[T] | 2 | 1.58E + 06 | 0 |
| [K].VREEVVTVGNSVNEG$^{2197}$.[L$^{2198}$]* | 1 | 1.56E + 06 | 0 |
| [G$^{2197}$].L$^{2198}$NQPTDDSCFDPYTVSHYAVGDEWER.[M]* | 1 | 6.82E + 06 | 0 |

Abbreviations used: PSM, Peptide spectrum matches.
*Peptides with termini identifying the same cleavage site (G$^{2197}$-L$^{2198}$).
$^†$Peptide unique to the FN1-8 isoform.

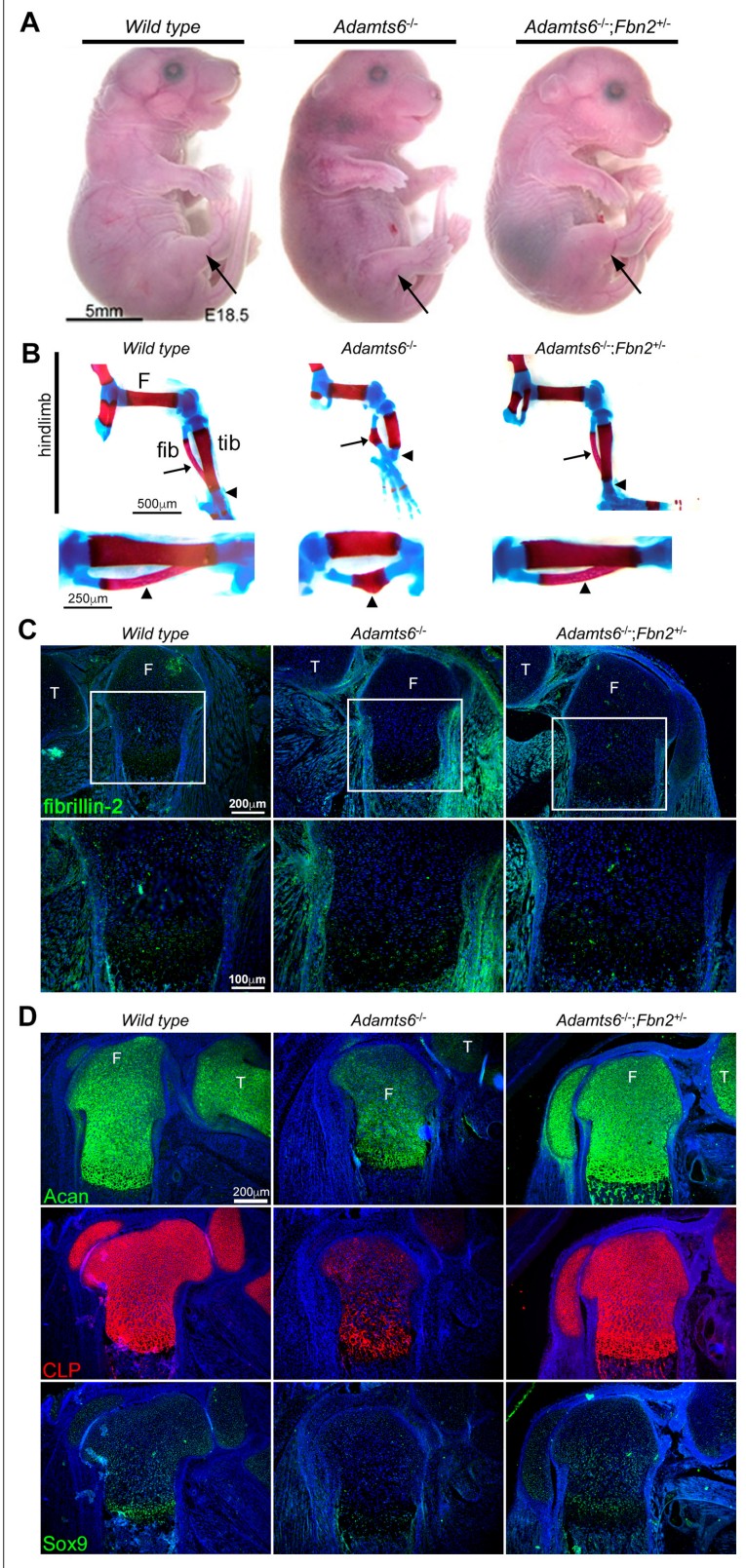

**Figure 8.** Genetic reversal of limb anomalies in *Adamts6* mutant mice by *Fbn2* haploinsufficiency. (**A–B**) Deletion of one *Fbn2* allele reverses limb dysmorphology in E18.5 *Adamts6*⁻/⁻ embryos, specifically, externally evident limb segment dimensions and reversal of rotational anomaly (arrow) (**A**), normal ossific centers and overall hindlimb skeletal structure, reversal of internal rotation and bending of the tibia (arrowheads) and restores normal tibial and

*Figure 8 continued on next page*

*Figure 8 continued*

fibular (arrows) length, shape and alignment (**B**). The lower panels in (**B**) illustrate the tibia and fibula (arrowhead) at higher magnification. (**C**) Fibrillin-2 staining is reduced to wild type levels in *Adamts6*$^{-/-}$;*Fbn2*$^{+/-}$ distal femoral perichondrium. T, tibia, F, femur. (**D**) Restitution of aggrecan, cartilage link protein (CLP) and Sox9 staining in *Adamts6*$^{-/-}$;*Fbn2*$^{+/-}$ distal femoral cartilage. T, tibia, F, femur.

The online version of this article includes the following figure supplement(s) for figure 8:

**Figure supplement 1.** Reversal of reduced body length and long bone shortening in *Adamts6*-deficient embryos by *Fbn2* hemizygosity.

**Figure supplement 2.** *Fbn2* haploinsufficiency reverses axial skeleton and craniofacial anomalies in *Adamts6*$^{-/-}$ mice.

**Figure supplement 3.** Normalized fibrillin-2, aggrecan, cartilage link protein and Sox9 staining intensity in *Adamts6*$^{-/-}$;*Fbn2*$^{+/-}$ cartilage.

**Figure supplement 4.** *Fbn2* hemizygosity does not itself alter hindlimb long bone length or growth plate morphology and dimensions.

**Figure supplement 5.** Genetic *Fbn1* reduction does not affect *Adamts6*$^{-/-}$ limb and skeletal defects.

---

*2008*). Thus, differential transcription of fibrillin genes favors reduced fibrillin-2 synthesis postnatally, and together with dominance of *Fbn1* expression, was thought to underlie the prevalence of fibrillin-1 microfibrils in juvenile and adult mice. Fibrillin-2 was identified postnatally in only a few locations, and in some tissues, microfibril bundles had a core of fibrillin-2 microfibrils surrounded by abundant fibrillin-1, suggesting that postnatal microfibrils (comprising fibrillin-1) were added on pre-existing fibrillin-2-rich microfibrils (*De Maria et al., 2017*; *Charbonneau et al., 2010*). Previous work also suggested that fibrillin-2 epitope availability was masked postnatally by such overlay of fibrillin-1. Specifically, it was shown that fibrillin-2 staining in postnatal tissues such as perichondrium was enhanced by digestion with collagenase, that microfibrils treated with chaotropic agent had enhanced reactivity to fibrillin-2 antibodies, and that *Fbn1* null juvenile mice, which die by 2 weeks of age, had robust fibrillin-2 staining (*Charbonneau et al., 2010*). Thus, it was thought that the fibrillin isoform content of microfibrils is substantially determined by the level of transcription of the respective genes and that fibrillin-2 fibrils are mostly masked postnatally, such that they may not have much of a regulatory role. The fate of fibrillin-2 microfibrils produced in the embryonic period, the possibility of specific proteolytic mechanisms to reduce their abundance, and a deleterious impact of fibrillin-2 overabundance on morphogenesis had not been previously considered and no animal models with excess fibrillin-2 are available.

The present work suggests that ADAMTS6 and ADAMTS10 work collaboratively, but in distinct ways, to support prevalence of fibrillin-1 in microfibrils after birth. ADAMTS10 is innately resistant to activation by furin, with only a small proportion converted to an active protease. ADAMTS10 promotes fibrillin-1 microfibril assembly (*Kutz et al., 2011*), consistent with the observation that recessive *ADAMTS10* mutations and dominantly inherited *FBN1* mutations each lead to Weill-Marchesani syndrome (*Dagoneau et al., 2004*; *Faivre et al., 2003a*). ADAMTS10 was therefore thought to function akin to several ADAMTS-like proteins, which lack catalytic activity, and accelerate biogenesis of fibrillin-1 containing microfibrils (*Hubmacher and Apte, 2015*; *Bader et al., 2010*; *Bader et al., 2012*; *Gabriel et al., 2012*; *Le Goff et al., 2008*; *Saito et al., 2011*; *Tsutsui et al., 2010*). ADAMTS10, if furin-activated by introduction of a furin cleavage site, was recently demonstrated to be proteolytically active against fibrillin-2 (*Wang et al., 2019a*), suggesting that the small fraction of ADAMTS10 that underwent activation could cleave fibrillin-2, and indeed, persistent fibrillin-2 fibrils were seen in *Adamts10*-deficient mice, which are viable to adulthood. The present work shows that ADAMTS6, whose zymogen is efficiently activated by furin (*Cain et al., 2016*), cleaves both monomeric fibrillin-1 and fibrillin-2, and has identified specific sites of cleavage of these substrates using TAILS and validated proteolysis biochemically in the case of the fibrillin-2 cleavage site. Fibrillin microfibril proteolysis was supported by loss of fibrillin-2 and fibrillin-1 microfibrils assembled by *Fbn1*$^{-/-}$ and *Fbn2*$^{-/-}$ fibroblasts, respectively, in the presence of active ADAMTS6. Although we cannot exclude the possibility that ADAMTS6 acts via activation of another protease in the cell culture system that was used, the direct binding of ADAMTS6 to the fibrillins in a binary interaction assay strongly supports direct cleavage. The significance of reduced fibrillin-2 proteolysis and fibrillin-2 accumulation after

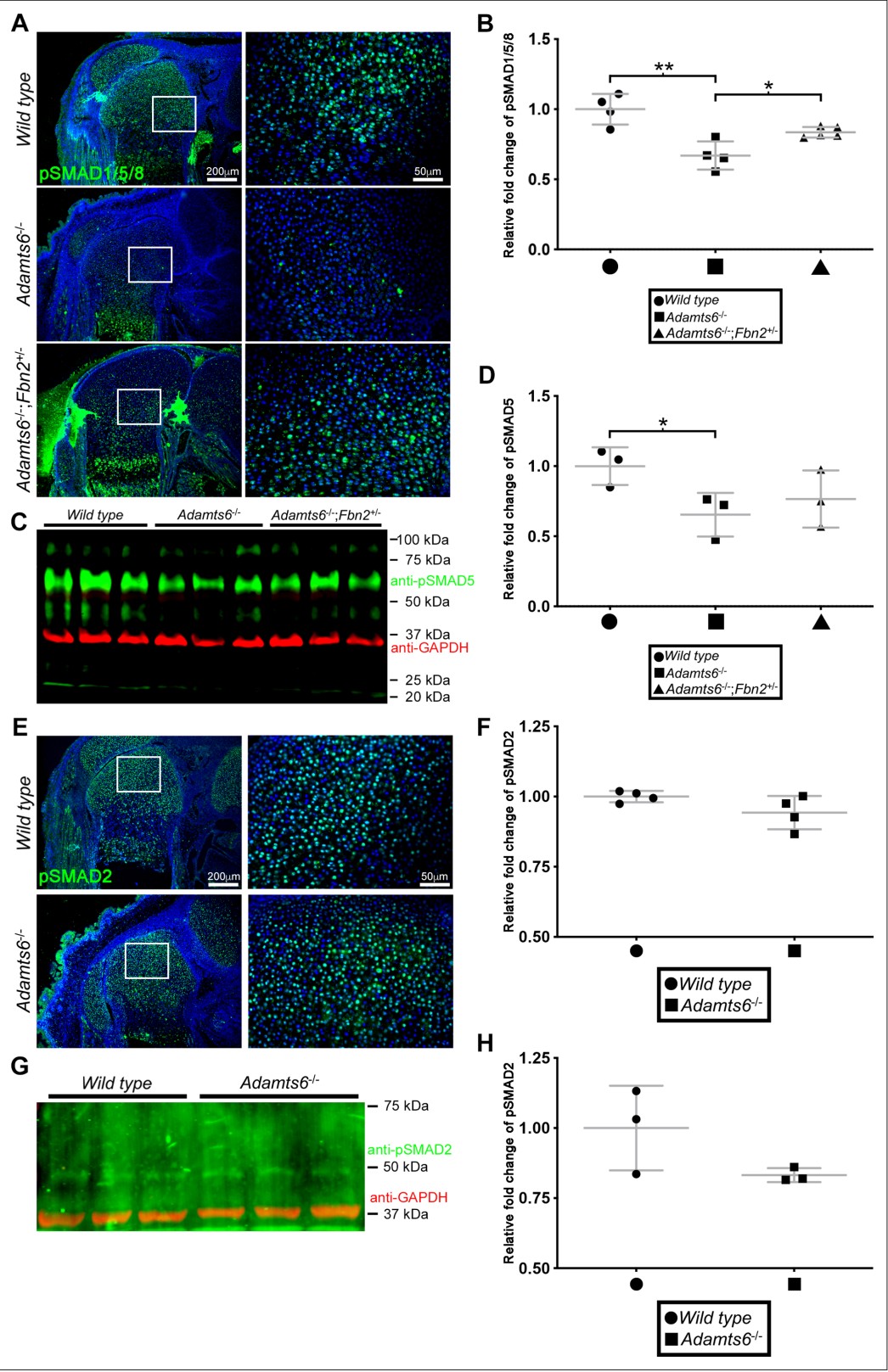

**Figure 9.** Reduced BMP signaling but unaltered canonical TGFβ signaling in *Adamts6^{-/-}* distal femur. (**A**) Reduced pSMAD1/5/8 staining in *Adamts6^{-/-}* femur, as compared to wild type control, is restored to wild type levels in *Adamts6^{-/-};Fbn2^{+/-}* femur as quantified in (**B**). n ≥ 4; *p ≤ 0.05; **p ≤ 0.01. (**C**) Western blotting of E18.5 hindlimb extracts shows reduced pSMAD5 (green; 58 kDa). Anti-GAPDH (red, 37 kDa) was used as a loading control. (**D**)

*Figure 9 continued on next page*

*Figure 9 continued*

pSMAD5 levels from the western blot in (**C**) was quantified after normalization to GAPDH loading control. n = 3. *p ≤ 0.05. (**E**) No change in pSMAD2 staining in *Adamts6*[-/-] femur as compared to wild-type control. The results are quantified in (**F**). (**G**) Western blot analysis shows no change in anti-pSMAD2 (green; 52 kDa) in *Adamts6*-deficient E18.5 hindlimb lysates. Anti-GAPDH (red, 37 kDa) was used as a loading control. (**H**) Quantification of anti-pSMAD2 signal in (**G**) using GAPDH signal intensity as the control. n = 3.

*Adamts6* inactivation was demonstrated by dramatic reversal of the observed skeletal and palate defects after introducing *Fbn2* haploinsufficiency in *Adamts6* mutants. The lack of a comparable effect after reduction of fibrillin-1 suggested a primacy of the ADAMTS6-fibrillin-2 relationship. Although ADAMTS6 cleaves fibrillin-1, it is not as essential for proteolysis of fibrillin-1, presumably because other proteases potentially compensate. While we cannot exclude the likelihood of accumulation of additional ADAMTS6 proteolytic targets in the skeleton, such as LTBP-1, which it was previously shown to cleave (*Cain et al., 2016*), the present work strongly suggests that fibrillin-2 proteolysis by ADAMTS6 is necessary for proper skeletogenesis.

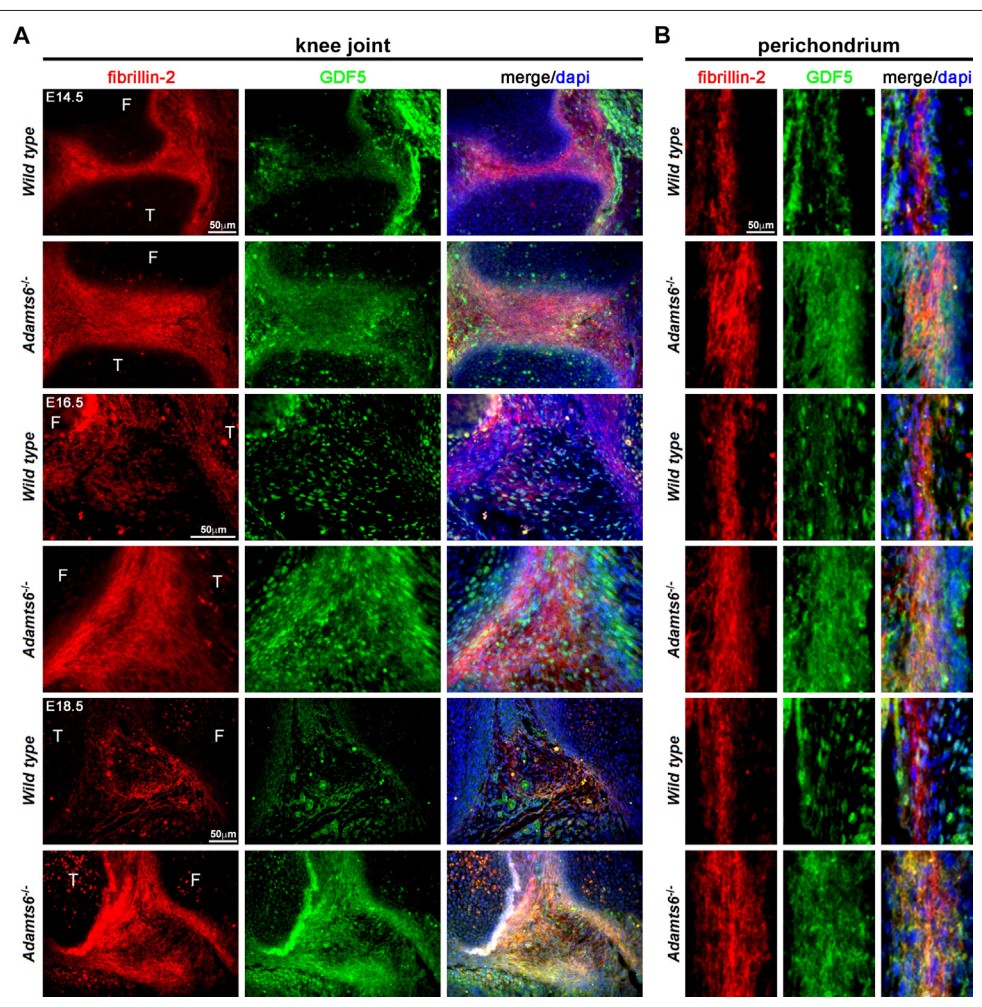

**Figure 10.** GDF5 co-localizes with fibrillin-2 microfibrils with greater staining intensity in *Adamts6*[-/-] knee joints. (**A–B**) E14.5, E16.5 and E18.5 *Adamts6*[-/-] knee joints (**A**) and perichondrium (**B**) show both increased fibrillin-2 (red) and GDF5 (green) staining in joint mesenchyme as compared to wild-type control. Note the overlapping staining of fibrillin-2 (red) and GDF5 (green) with increased immunodetection of both fibrillin-2 and GDF5 in the *Adamts6*[-/-] knee joint mesenchyme. F, femur; T, tibia.

The online version of this article includes the following figure supplement(s) for figure 10:

**Figure supplement 1.** Increased fibrillin-2 and GDF5 staining intensity in *Adamts6*[-/-] knee joint mesenchyme.

In addition to mechanical effects of excess perichondrial fibrillin-2 on cartilage that could affect skeletal development, reduced fibrillin-2 proteolysis could lead to dysregulated sequestration and release of growth factors of the TGFβ superfamily, since fibrillin-2 can bind to numerous superfamily members including several BMPs and GDFs (*Sengle et al., 2015*; *Gregory et al., 2005*; *Sengle et al., 2011*). Fibrillin microfibrils appear to have little role in activation of the BMPs, and primarily affect their local concentrations and release, for example, by BMP cleavage, which was recently attributed to MMPs (*Furlan et al., 2021*; *Wohl et al., 2016*). In contrast, proteolysis of fibrillins was not previously implicated in control of local BMP/GDF activity. In the present work, the microfibril changes in the perichondrium were associated with changes in the underlying cartilage, in further support of phenotype modulation across the perichondrium-cartilage boundary (*Colnot et al., 2005*; *Maes, 2017*; *Vortkamp et al., 1996*). Prior work has shown that combinations of TGFβ superfamily members strongly promote expression of typical cartilage genes (*Murphy et al., 2015*; *Hatakeyama et al., 2004*). This previous work together with the observed reduction in pSmad5 levels (indicative of reduced BMP signaling) may explain reduced Sox9, aggrecan and link protein that we observed. The resulting reduced cartilage proteoglycan potentially results in softer cartilage that could be susceptible to angulation forces as limb muscles develop.

Consistent with the conclusion that *Adamts6* deficiency reduces limb cartilage BMP signaling as a result of excess fibrillin-2, *Fbn2*-deficent mice have increased BMP signaling (*Sengle et al., 2015*). Thus, accumulation of fibrillin-2 microfibrils resulting from combined ADAMTS6 and ADAMTS10 deficiency may excessively sequester BMPs and GDFs in the perichondrium, leading to impaired BMP signaling in the underlying cartilage (**Figure 11**). Fibrillins interact with the related latent TGFβ-binding proteins and furthermore, MAGP1, a fibrillin-binding partner shown to accumulate in the *Adamts6* mutant mice, also regulates the cellular microenvironment via growth factor binding (*Miyamoto et al., 2006*; *Nehring et al., 2005*). Moreover, ADAMTS6-mediated proteolysis of cell-surface heparan-sulfate proteoglycans (*Cain et al., 2016*) could also influence the skeletal microenvironment. However, although such complex effects on diverse soluble morphogens and growth factors may ensue downstream of the observed ECM changes, the genetic evidence from our studies unequivocally indicates that fibrillin-2 accumulation is a central mechanism underlying the skeletal defects observed in the *Adamts6* mutant limbs. Thus, the present work highlights the significance of specific proteolytic mechanisms in fibrillin microfibril proteostasis, their importance for morphogen/growth factor regulation by ECM and ultimately, for cellular regulation by ECM.

# Materials and methods

## Key resources table

| Reagent type (species) or resource | Designation | Source or reference | Identifiers | Additional information |
|---|---|---|---|---|
| Gene (*Mus musculus*) | *Adamts6* | Genbank | NCBI Gene: 108,154 | ADAM metallopeptidase with thrombospondin type 1 motif 6 |
| Gene (*Mus musculus*) | *Adamts10* | Genbank | NCBI Gene: 224,697 | ADAM metallopeptidase with thrombospondin type 1 motif 10 |
| Gene (*Mus musculus*) | *Adamts17* | Genbank | NCBI Gene: 233,332 | ADAM metallopeptidase with thrombospondin type 1 motif 17 |
| Gene (*Mus musculus*) | *Adamts19* | Genbank | NCBI Gene: 240,322 | ADAM metallopeptidase with thrombospondin type 1 motif 19 |
| Gene (*Mus musculus*) | *Fbn1* | Genbank | NCBI Gene: 14,118 | Fibrillin-1 |
| Gene (*Mus musculus*) | *Fbn2* | Genbank | NCBI Gene: 14,119 | Fibrillin-2 |
| Gene (*Mus musculus*) | *Fn1* | Genbank | NCBI Gene: 14,268 | Fibronectin |
| Strain, strain background (*Mus musculus*) | *Adamts6*[b2b2029Clo] (C57BL/6 J) | Mouse Genome Informatics | RRID: MGI:5487397 | *Adamts6* null allele |
| Strain, strain background (*Mus musculus*) | *Adamts10*[tm1Dgen] (C57BL/6 J) | Mouse Genome Informatics | RRID: MGI:6355992 | *Adamts10* null allele |
| Strain, strain background (*Mus musculus*) | *Fbn1*[tm2Rmz] (C57BL/6 J) | Mouse Genome Informatics | RRID: MGI:1934906 | *Fbn1*[mgR] allele |

*Continued on next page*

Continued

| Reagent type (species) or resource | Designation | Source or reference | Identifiers | Additional information |
|---|---|---|---|---|
| Strain, strain background (*Mus musculus*) | *Fbn2*<sup>tm1Rmz</sup> (C57BL/6 J) | Mouse Genome Informatics | RRID: MGI:3652417 | *Fbn2* null allele |
| Strain, strain background (*Mus musculus*) | *Fbn1*<sup>tm3Rmz</sup> (C57BL/6 J) | PMCID:PMC3875392 | RRID: MGI:3641232 | *Fbn1* null allele MEFs used for cell culture experiments |
| Strain, strain background (*Mus musculus*) | *Fbn2*<sup>tm1Rmz</sup> (C57BL/6 J) | This paper | RRID: MGI:3652417 | *Fbn2* null MEFs used for in vitro microfibril staining |
| Strain, strain background (*Mus musculus*) | C57BL/6 J strain | This paper | RRID: IMSR_JAX:000664 | Wild type MEFs used for in vitro microfibril staining |
| Strain, strain background (*Mus musculus*) | *Adamts6*<sup>b2b2029Clo</sup> (C57BL/6 J) | This paper | RRID: MGI:5487397 | *Adamts6* mutant MEFs for in vitro microfibril staining |
| Transfected construct (*Mus musculus*) | ADAMTS6 WT | PMCID:PMC6048820 | | ADAMTS6 plasmid |
| Transfected construct (*Mus musculus*) | ADAMTS6 EA | This paper | This paper | Catalytically- inactive ADAMTS6 plasmid |
| Cell line (human) | FBN1-expressing cells | Dieter Reinhardt, Ph.D. | PMID:12399449 | Stable HEK293 cell line expressing full length FBN1 |
| Cell line (human) | FBN2-Nt-expressing cells | Dieter Reinhardt, Ph.D. | PMID:12399449 | Stable HEK293 cell line expressing the N-terminal half of FBN2 |
| Cell line (human) | FBN2-Ct-expressing cells | Dieter Reinhardt, Ph.D. | PMID:12399449 | Stable HEK293 cell line expressing the C-terminal half of FBN2 |
| Peptide, recombinant protein | FBN2-Nt-expressing cells | Dieter Reinhardt, Ph.D. | rFBN2-N; PMID:12399449 | Used for Biacore analysis |
| Peptide, recombinant protein | FBN2-Ct | Dieter Reinhardt, Ph.D. | rFBN2-C; PMID:12399449 | Used for Biacore analysis |
| Peptide, recombinant protein | FBN1-Nt | Dieter Reinhardt, Ph.D. | rFBN1-N; PMID:12399449 | Used for Biacore analysis |
| Peptide, recombinant protein | FBN1-Ct | Dieter Reinhardt, Ph.D. | rFBN1-C; PMID:12399449 | Used for Biacore analysis |
| Peptide, recombinant protein | FN1 | Deane Mosher, M.D. | PMD:6133865 | Used for Biacore analysis |
| Peptide, recombinant protein | ADAMTS6-Ct | Stuart Cain, Ph.D. | PMCID:PMC5078793 | Used for Biacore analysis |
| Peptide, recombinant protein | ADAMTS6-4TSR | Stuart Cain, Ph.D. | This paper | Used for Biacore analysis |
| Peptide, recombinant protein | ADAMTS6-S4TSR | Stuart Cain, Ph.D. | This paper | Used for Biacore analysis |
| Peptide, recombinant protein | ADAMTS6-TCS | Stuart Cain, Ph.D. | This paper | Used for Biacore analysis |
| Antibody | Anti-Fibrillin-2-Gly (rabbit polyclonal) | Robert Mecham, Ph.D. | PMID:10825173 | IF (1:300) |
| Antibody | Anti-mFbn1-C (rabbit polyclonal) | Dieter Reinhardt, Ph.D. | PMID:33039488 | IF (1:500) |
| Antibody | Anti-rFBN2-C (rabbit polyclonal) | Dieter Reinhardt, Ph.D. | PMID:12399449 | WB (1:500) |
| Antibody | Anti-MAGP1 (rabbit polyclonal) | Robert Mecham, Ph.D. | PMCID:PMC14862 | IF (1:200) |
| Antibody | Anti-Sox9 (rabbit polyclonal) | Millipore AB5535 | RRID: AB_2239761 | IF (1:300) |
| Antibody | Anti-Acan (rabbit polyclonal) | Millipore AB1031 | RRID: AB_90460 | IF (1:400) |
| Antibody | Anti-CLP (mouse monoclonal) | DSHB 9/30/8 A-4 | RRID: AB_2248142 | IF (1:100) |
| Antibody | Anti-Col X (rabbit polyclonal) | Abcam ab58632 | RRID: AB_879742 | IF (1:1000) |
| Antibody | Anti-PCNA (mouse monoclonal) | Cell Signaling 2,586 | RRID: AB_2160343 | IF (1:200) |
| Antibody | Anti-His (mouse monoclonal) | R&D MAB050 | RRID: AB_357353 | IF (1:400) WB (1:1000) |

*Continued*

| Reagent type (species) or resource | Designation | Source or reference | Identifiers | Additional information |
|---|---|---|---|---|
| Antibody | Anti-pSmad5 (rabbit polyclonal) | Abcam ab92698 | RRID: AB_10561456 | WB (1:1000) |
| Antibody | Anti-pSmad1/5 (rabbit polyclonal) | Cell Signaling 9,516 | RRID: AB_491015 | IF (1:200) |
| Antibody | Anti-pSmad2 (rabbit polyclonal) | Cell Signaling 3,108 | RRID: AB_490941 | IF (1:200) WB (1:1000) |
| Antibody | Anti-GDF5 (goat polyclonal) | R&D AF853 | RRID: AB_355662 | IF (1:100) |
| Antibody | Anti-GAPDH (mouse monoclonal) | Millipore MAB374 | RRID: AB_2107445 | WB (1:5000) |
| Commercial assay or kit | *Adamts6* RNAscope probe | ACD Bio | 428,301 | Mouse ISH probe |
| Commercial assay or kit | *Adamts10* RNAscope probe | ACD Bio | 585,161 | Mouse ISH probe |
| Commercial assay or kit | RNAscope 2.5 HD Red in situ detection kit | ACD Bio | 322,350 | Used to detect *Adamts6* and *Adamts10* probes |
| Software, algorithm | GraphPad Prism | GraphPad | RRID: SCR_002798 | Utilized for statistical computing of data |
| Software, algorithm | Fiji | NIH | RRID: SCR_002285 | Used to quantify IF tissue sections |

IF = Immunofluorescence.. WB = western blot.. ISH = in situ hybridization.

## Transgenic mice

*Adamts6*[b2b2029Clo] (RRID:MGI:5487287), *Adamts10*[tm1Dgen] (RRID:MGI:6355992), *Fbn1*[mgR] (*Fbn1*[tm2Rmz]; RRID:MGI:1934906) and *Fbn2*[tm1Rmz] (RRID:MGI:3652417) mice were previously described (*Arteaga-Solis et al., 2001*; *Wang et al., 2019a*; *Prins et al., 2018*; *Pereira et al., 1999*; *Li et al., 2015*) and maintained on a C57BL/6 J background. All mouse experiments were approved by the Cleveland Clinic Institutional Animal Care and Use Committee (protocol 2018–2450).

## Histology and immunofluorescence microscopy

Forelimbs and hindlimbs were fixed with 4% paraformaldehyde (PFA) in PBS at 4 °C for 48 hr. 7 µm sections were used for histochemistry (alcian blue or RGB trichrome stain *Gaytan et al., 2020*) or for indirect immunofluorescence. The primary antibodies (*Supplementary file 1*) were followed by secondary goat anti-mouse or goat anti-rabbit antibody (A11004 or A11008; 1:400; Invitrogen, Thousand Oaks, CA) treatment. Prior to immunofluorescence, citrate antigen retrieval, immersion of slides in citrate-EDTA buffer (10 mM/l citric acid, 2 mM/l EDTA, 0.05% v/v Tween-20, pH 6.2) and microwaving for 4 intervals of 1.5 min at 50% power in a microwave oven with 30 s intervals between heating cycles, was utilized. In addition, hyaluronidase treatment of sections (0.2% in PBS; H-2251; Sigma) was used prior to fibrillin-2 and fibrillin-1 immunostaining. After depariffinization and rehydration, sections were stained with 1% alcian blue 8 GX (A3157; Sigma) in 3% acetic acid (pH 2.5) for 10 min, rinsed in tap water, counter-stained in nuclear fast red (J61010; Alfa Aesar) for 1 min, rinsed in tap water prior to dehydration and mounted as previously described (*Mead and Yutzey, 2009*). RGB-trichrome staining was done essentially as previously described (*Gaytan et al., 2020*). Briefly, after de-paraffinization and rehydration, the sections were stained with 1% alcian blue 8 GX (A3157; Sigma) in 3% acetic acid (pH 2.5) for 20 min, rinsed in tap water, 1% fast green FCF (F-99; Fisher Scientific) for 20 min, rinsed in tap water, 0.1% sirius red in saturated picric acid (26357–02; EMS) for 60 min, rinsed in 2 changes of 1% acetic acid prior to dehydration in 100% ethanol and xylene. TUNEL assay (11684795910; Sigma) was performed as described previously (*Gaytan et al., 2020*). Images were obtained using an Olympus BX51 microscope with Leica DFC 7000T camera and Leica Application Suite V4.6 software (Leica, Wetzlar, Germany). Histological sections were masked during data collection and quantified utilizing NIH Fiji software (*Schindelin et al., 2012*).

Alizarin red-alcian blue stained skeleton preparations were performed as described (*Mead, 2020a*). Briefly, skinned, eviscerated mice were fixed in 80% ethanol for 24 hr, dehydrated in 95% ethanol for 24 hr and acetone for 48 hr, stained (0.1% alizarin red S, 0.3% alcian blue, 1% glacial acetic acid in 95% ethanol) for 48 hr, cleared in 95% ethanol for 1 hr, muscle tissue was gently removed with forceps, and the preparations were cleared in a series of increasing (20–80%) glycerol/1% KOH ratio until storage

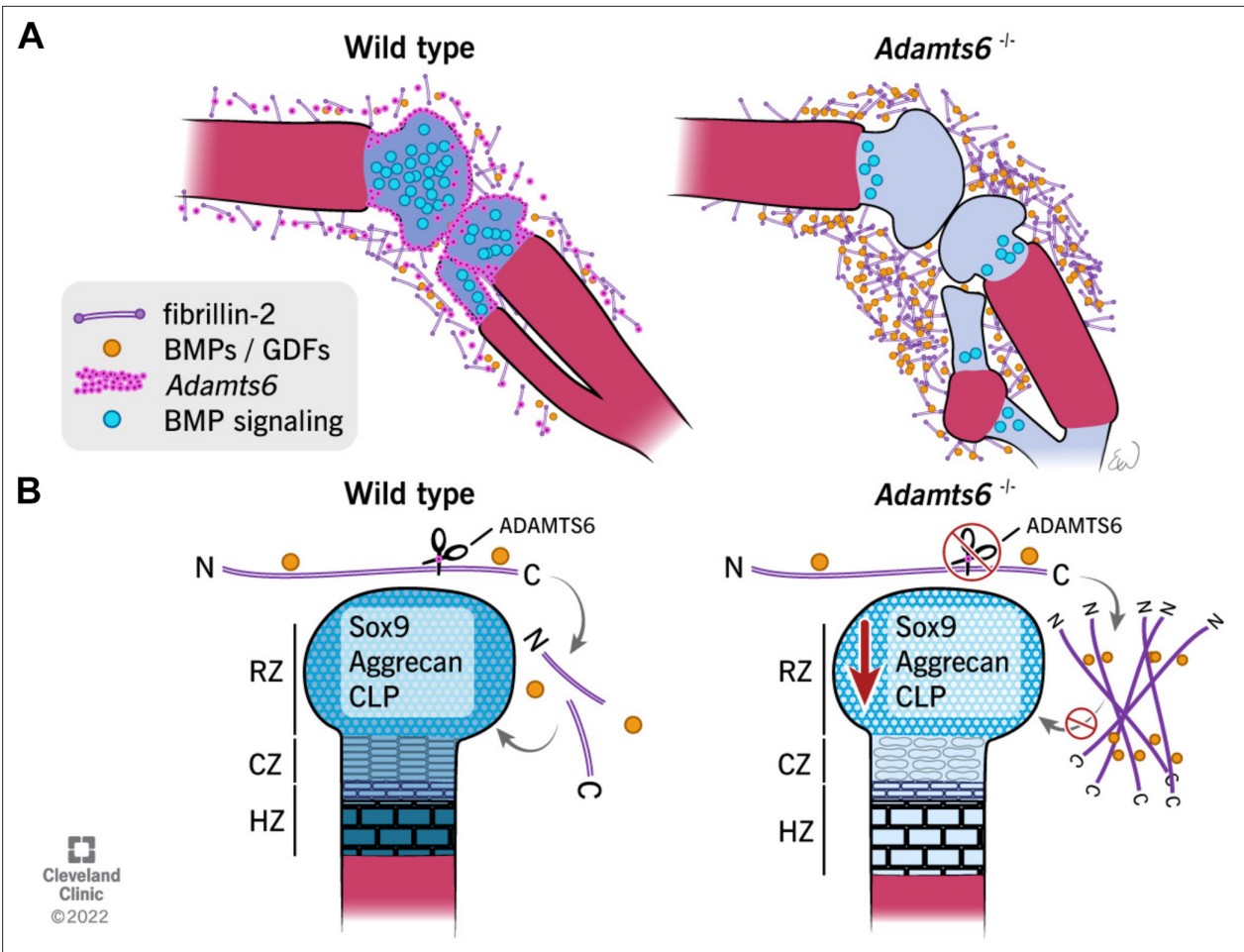

**Figure 11.** Schematic illustrating the observed anomalies in the absence of ADAMTS6 and proposed roles and mechanisms of ADAMTS6 in skeletal development. (**A**) *Adamts6* expression (pink dots) is shown in the perichondrium, peripheral chondrocytes, and in mesenchyme surrounding the knee joint. BMP signaling (blue circles) is evident in growth plate cartilage while fibrillin-2 (purple rods) and BMP/GDF (orange dots) are co-localized in the mesenchyme surrounding the knee joint and long bones. *Adamts6*-deficient limbs have a reduction of BMP signaling in the growth plate, retention of fibrillin-2 microfibrils and sequestration of GDF5, and presumably, other known fibrillin-2 binding BMPs/GDFs. (**B**) The schematic shows that in wild type limbs, ADAMTS6 cleaves fibrillin-2, allowing release of growth factors and contributing to proper cartilage growth and development. In the absence of ADAMTS6, there is an over-abundance of fibrillin-2 microfibrils, sequestration of GDF5/BMPs and as a consequence, reduced Sox9, aggrecan (blue shading) and cartilage link protein, impairing cartilage differentiation and structural integrity. Mechanistically, we conclude from the data that ADAMTS6 acts principally via fibrillin-2 cleavage and release of sequestered BMP/GDF. RZ, resting zone; CZ, columnar zone; HZ, hypertrophic zone; N, N-terminus; C, C-terminus of fibrillin-2.

in 100% glycerin and photography (Leica MZ6; Insight Spot software camera and software). Crown to rump or individual bone length measurements were masked during data collection utilizing NIH Fiji software (*Schindelin et al., 2012*).

## RNA in situ hybridization (ISH)

*Adamts6* and *Adamts10* ISH was performed using RNAscope (Advanced Cell Diagnostics, Newark, CA) as described (*Mead and Apte, 2020b*). Briefly, 6 µm sections were deparaffinized and hybridized to mouse *Adamts6* and *Adamts10* probe sets (428301, 585161, respectively; Advanced Cell Diagnostics) using a HybEZ oven (Advanced Cell Diagnostics) and the RNAScope 2.5 HD Detection Reagent Kit (322360; Advanced Cell Diagnostics) and counterstained with eosin.

## Cell culture

HEK293F cells were purchased and authenticated from ATCC and maintained in Dulbecco's Modified Eagle Medium (DMEM) supplemented with 10% fetal bovine serum (FBS), 100 U/ml penicillin and 100 µg/ml streptomycin at 37 °C in 5% $CO_2$ using a humidified chamber. The cells were transiently transfected with ADAMTS6 WT or ADAMTS6 EA expression plasmids using PEI MAX (24765; Polysciences) and were co-cultured with $Fbn1^{-/-}$ mouse embryo fibroblasts MEFs, prepared from $Fbn1^{tm3Rmz}$ homozygous mutants (*Carta et al., 2006*; *Beene et al., 2013*) or $Fbn2^{-/-}$ mouse skin fibroblasts (MSFs) (*Beene et al., 2013*) in a 1:1 ratio on 8-well culture slides (354118; Falcon). Similarily, $Adamts6^{-/-}$ and wild type MEFs were plated on 8-well culture slides for immunofluorescent staining. The cells were cultured for 6 days, fixed in ice-cold 70% methanol/30% acetone for 5 min at room temperature, blocked with 5% normal goat serum in PBS for 1 hr at room temperature and incubated with primary antibody (*Supplementary file 1*) overnight at 4 °C as described (*Hubmacher et al., 2017*). The cells were washed three times with PBS for 5 min at room temperature and incubated with Alexa-Fluor labeled secondary antibodies (goat anti-mouse 568 or goat anti-rabbit 488; Invitrogen A11004, A11008, respectively, 1:400).

## RNA isolation and quantitative real-time PCR (qRT-PCR)

Mouse hindlimbs, hearts and lungs were snap-frozen and stored at –80 °C until use. Total RNA was isolated using TRIzol (15596018, Invitrogen), and 2 µg of RNA was reverse transcribed into cDNA using a High-Capacity cDNA reverse transcription kit following the manufacturer's instructions (4368814; Applied Biosystems, Foster City, CA). qRT-PCR was performed with Bullseye EvaGreen qPCR MasterMix (BEQPCR-S; MIDSCI) using an Applied Biosystems 7,500 instrument. The experiments were performed with three independent biological samples and reproducibility was confirmed with two to three technical replicates. *Gapdh* was used as a control for mRNA quantity. The ΔΔCt method was used to calculate relative mRNA expression levels of target genes and shown as standard error of the mean (SEM). See *Supplementary file 2* for primer sequences.

## Surface plasmon resonance analysis

The human fibrillin-2 recombinant halves (rFBN2-N and rFBN2-C, termed FBN2-Nt and FBN2-Ct in this manuscript, respectively) were purified to homogeneity (> 90% purity) as described previously (*Lin et al., 2002*). Recombinant human ADAMTS6-Ct (residues 558–1117), ADAMTS6-S4TSR (residues 717–1117), ADAMTS6-4TSR (residues 840–1117) and ADAMTS6-TCS (residues 558–839) were expressed using the mammalian expression vector pCEP-pu/AC7 modified with a C-terminal V5-His6 tag, in 293-EBNA cells as described previously (*Cain et al., 2016*). Purified FBN2-Ct or ADAMTS6-Ct in 10 mM acetate, pH 4.0 were immobilized on a Biacore CM5 sensor chip (research grade) with the amine coupling kit following the manufacturer's instructions (GE Healthcare). A total of 1700 resonance units of FBN2-Ct or ADAMTS6-Ct was coupled to the chip for analysis in a Biacore 3000 instrument (GE Healthcare). The kinetics analysis was performed at 25 °C in 10 mM Hepes buffer, pH 7.4 with 0.15 M NaCl, 2 mM $CaCl_2$, and 0.005% or 0.05% (v/v) surfactant P20 at a flow rate of 30 µl/min. All the analytes were diluted in the above buffer at different concentrations and injected through an uncoupled control flow cell in series with the flow cell coupled with FBN2-Ct or ADAMTS6-Ct constructs. The sample injection time was 2 min for ADAMTS6 and 3 min for FBN2 analytes. The dissociation time was 10 min. 1 M ethanolamine, pH 8.5 was used for chip surface regeneration at a flow rate of 30 µl/min for 30–60 s followed by 2 min stabilization time. All data were corrected with reference to the background binding in the control flow cell. The association and disassociation curves were generated with the BIAevaluation software (version 4.0.1; GE Healthcare). The kinetic constants were calculated using the steady state affinity method. Similar experiments were performed with the human fibrillin-1 recombinant halves (rFBN1-N and rFBN1-C, termed FBN1-Nt and FBN1-Ct in this manuscript, respectively) (*Lin et al., 2002*) with ADAMTS6-Ct.

## Site-directed mutagenesis, transient transfection, and western blotting

A plasmid encoding mouse ADAMTS6 with a C-terminal myc/his tag was generated previously (*Prins et al., 2018*) and used for site-directed mutagenesis (Q5 Site-Directed Mutagenesis Kit; E0554; New England BioLabs) to introduce Ala at Glu[404], a classic metalloprotease inactivating mutation (ADAMTS6 EA). Plasmids were transfected into HEK293F cells using PEI MAX (24765; Polysciences)

and conditioned medium was collected 48–72 hr later. Aliquots of medium were analyzed by 7.5% reducing SDS-PAGE. Proteins were electroblotted to polyvinylidene fluoride membranes (IPFL00010, EMD Millipore, Billerica, MA), incubated with primary antibodies (*Supplementary file 1*) overnight at 4 °C, followed by fluorescent secondary antibody goat anti mouse or rabbit (827–08365, 926–68170; 1:5000; Li-COR Biosciences, Lincoln, NE) for 1 hr at room temperature. Antibody binding was visualized using an ODYSSEY CLx infrared imaging system (LI-COR). For pSMAD2 and pSMAD5 detection, hindlimbs were placed in RIPA Buffer (ab156034; Abcam) and Complete Protease Inhibitor Cocktail (no 4693159001; Millipore) and PhosSTOP (Millipore no. 4906845001) were added prior to homogenization T10 basic ULTRA-TURRAX (IKA, Staufen, Germany) and ultrasonication (Qsonica, Newtown, CT, USA)(3 × 2 s at 20% with 3 s pause). The supernatant was collected after centrifugation and 100 µg loaded on a 10% gel. Western blot band intensity was quantified utilizing NIH Fiji software (*Schindelin et al., 2012*).

## Cell co-culture for degradomics
Transiently transfected HEK293F cells expressing wild type ADAMTS6 or ADAMTS6 EA were cocultured with HEK293F cells stably expressing either fibrillin-2 Ct or full-length fibrillin-1. To determine whether fibronectin was an ADAMTS6 substrate, HEK 293F cells were transiently transfected with either ADAMTS6 or ADAMTS6 EA plasmids and seeded onto wells coated with human fibronectin as previously described (*Wang et al., 2019b*). These experiments used serum-free and phenol red-free DMEM.

## TAILS sample workflow
The conditioned medium from the above cell cultures were centrifuged at 4,000 rcf for 20 min at 4 °C and the supernatant was filtered through a 0.22 µM filter. The medium was concentrated 20-fold using a 3 kDa (Amicon) stirring filter. Proteins were isolated using chloroform/methanol precipitation and resuspended in 2.5 M GuHCl and 250 mM HEPES pH 7.8. Protein concentration was measured using the Bradford assay (Pierce, Thermo) to determine the volume needed for 500 µg of protein from each condition. Proteins were reduced with 10 mM dithiothreitol (DTT) for 30 min at 37 °C followed by alkylation with 20 mM N-ethylmaleimide in the dark for 20 min. The reaction was quenched by adding DTT to a final concentration of 20 mM. Proteins were labeled overnight with 40 mM light or heavy formaldehyde, which binds specifically to free N-termini and lysine residues (α and ε amines, respectively) in the presence of 20 mM sodium cyanoborohydride at 37 °C as described (*Martin et al., 2020*). They were treated with an additional fresh 20 mM formaldehyde and 10 mM sodium cyanoborohydride for 2 hr the following day at 37 °C and the reaction was quenched with 100 mM Tris for 1 hr at 37 °C. 500 µg of each sample was combined for buffer exchange on a 3 kDa molecular weight cut-off column (EMD Millipore) into 100 mM ammonium bicarbonate and digested overnight at 37 °C with mass spectrometry grade trypsin gold (Promega) at a 1:50 trypsin:protein ratio. Peptides were eluted via centrifugation and 30 µg of this digest was reserved for pre-TAILS analysis. The remaining peptides underwent enrichment using hyperbranched polyglycerol-aldehyde polymers (HPG-ALD, Flintbox, https://www.flintbox.com/public/project/1948/) at a 5:1 polymer:protein ratio. HPG-ALD binds to unblocked (trypsin-generated) amino acid termini, excluding them from the sample and thus enriches for blocked/labeled N-termini (*Kockmann et al., 2016*). The peptides were filtered through a 10 kDa MWCO column (EMD Millipore) to remove the polymer and obtain the TAILS fraction. TAILS and pre-TAILS fractions were desalted on a C18 Sep-Pak column (Waters) and eluted in 60:40 ACN: 1% trifluoroacetic acid. Samples were vacuum-centrifuged until dry and resuspended in 1% acetic acid for mass spectrometry.

## Mass spectrometry
Samples were analyzed on a ThermoFisher Scientific Fusion Lumos tribrid mass spectrometer system with a Thermo Ultimate 3,000 UHPLC interface. The HPLC column was a Dionex 15 cm x 75 µm internal diameter Acclaim Pepmap C18, 2 µm, 100 Å reversed- phase capillary chromatography column. Five µL volumes of the samples were injected and the peptides eluted from the column by an acetonitrile/0.1% formic acid gradient at a flow rate of 0.3 µL/min were introduced into the source of the mass spectrometer on-line over a 120 min gradient. The nanospray ion source was operated at 1.9 kV. The digest was analyzed using a data-dependent method with 35% collision-induced dissociation

fragmentation of the most abundant peptides every 3 s and an isolation window of 0.7 m/z for ion-trap MS/MS. Scans were conducted at a maximum resolution of 120,000 for full MS. Dynamic exclusion was enabled with a repeat count of 1 and ions within 10 ppm of the fragmented mass were excluded for 60 s.

### Proteomics data analysis

Peptides were identified using a precursor mass tolerance of 10 ppm, and fragment mass tolerance of 0.6 Da. The only static modification was carbamidomethyl (C), whereas dynamic modifications included the light (28.03 Da) dimethyl formaldehyde (N-terminal, K), heavy (34.06) dimethyl formaldehyde (N-terminal, K), oxidation (M, P), deamidation (N), acetylation (N-terminal), and Gln to pyro-Glu N-terminal cyclization. Peptides were validated using a false discovery rate (FDR) of 1% against a decoy database. Only high-confidence proteins (containing peptides at a 99% confidence level or higher) were recorded from each sample for data analysis. Internal peptides were identified based on the criteria of having an N-terminal modification, and a sequence that does not begin prior to the third amino acid in the protein or immediately following a known signal, transit, or propeptide sequence. Peptides that met these criteria were further analyzed based on the average fold-change ratio (ADAMTS6 WT/ ADAMTS6 EA sample abundance) between the three technical replicate pairs. The internal peptide abundance was divided by the total protein abundance fold-change to account for differences in protein levels between groups. Peptides that met these criteria and contained a weighted ratio (ADAMTS6 WT/ ADAMTS6 EA) greater than 1 underwent a t-test for significance. The mass spectrometry proteomics data have been deposited to the ProteomeXchange Consortium via the PRIDE partner repository with the dataset identifier PXD027096 and 10.6019/PXD027096.

### Statistics

Representative data of three independent, biological replicates are reported unless otherwise indicated. The two-tailed, unpaired Student t-test was used to obtain p values. Asterisks indicate differences with statistical significance as follows: $*p \leq 0.05$; $**p \leq 0.01$; $***p \leq 0.001$. In the dimethyl-TAILS experiments a two-tailed, paired Student t-test was used to obtain p values. Asterisks indicate differences with statistical significance as follows: $*p \leq 0.01$, $\# \leq 0.05$.

## Acknowledgements

This work was supported by the Allen Distinguished Investigator Program, through support made by The Paul G Allen Frontiers Group and the American Heart Association (to SSA), the National Institutes of Health (F32AR063548 and RO1HL156987 to TJM) and the David and Lindsay Morgenthaler Postdoctoral Fellowship (to TJM). DPR was supported by the Canadian Institutes of Health Research (PJT-162099) and the Marfan Foundation (USA). The Wellcome Centre for Cell-Matrix Research is supported by funding from Wellcome (203128/Z/16/Z). CB gratefully acknowledges BBSRC funding (Ref: BB/R008221/1). We thank Dr. Francesco Ramirez for *Fbn1* and *Fbn2* mutant mice, Dr. Deane Mosher for fibronectin and Dr. Robert Mecham for fibrillin-2 antibody.

## Additional information

### Funding

| Funder | Grant reference number | Author |
| --- | --- | --- |
| American Heart Association | 17DIA33820024 | Suneel S Apte |
| National Institutes of Health | F32AR063548 | Timothy J Mead |
| National Institutes of Health | RO1HL156987 | Timothy J Mead |
| Canadian Institutes of Health Research | PJT-162099 | Dieter Reinhardt |

| Funder | Grant reference number | Author |
|---|---|---|
| Marfan Foundation | | Dieter Reinhardt |
| Wellcome Trust | 203128/Z/16/Z | Clair Baldock |
| Biotechnology and Biological Sciences Research Council | BB/R008221/1 | Clair Baldock |
| Paul G Allen Frontiers Group | | Suneel S Apte |

The funders had no role in study design, data collection and interpretation, or the decision to submit the work for publication.

### Author contributions

Timothy J Mead, Conceptualization, Data curation, Formal analysis, Investigation, Methodology, Writing - original draft, Writing - review and editing; Daniel R Martin, Conceptualization, Data curation, Investigation, Methodology, Writing - review and editing; Lauren W Wang, Data curation, Formal analysis, Investigation, Writing - review and editing; Stuart A Cain, Formal analysis, Resources, Writing - review and editing; Cagri Gulec, Investigation, Writing - review and editing; Elisabeth Cahill, Joseph Mauch, Formal analysis, Investigation; Dieter Reinhardt, Cecilia Lo, Clair Baldock, Resources, Writing - review and editing; Suneel S Apte, Conceptualization, Formal analysis, Funding acquisition, Project administration, Supervision, Writing - original draft, Writing - review and editing

### Author ORCIDs

Timothy J Mead http://orcid.org/0000-0003-1891-3652
Suneel S Apte http://orcid.org/0000-0001-8441-1226

### Ethics

All of the animals were handled according to approved institutional animal care and use committee (IACUC) protocol 2018-2450 at the Cleveland Clinic.

### Decision letter and Author response

Decision letter https://doi.org/10.7554/eLife.71142.sa1
Author response https://doi.org/10.7554/eLife.71142.sa2

## Additional files

### Supplementary files

- Transparent reporting form
- Source data 1. Original, unedited western blots.
- Supplementary file 1. Antibodies.
- Supplementary file 2. Quantitative Real-Time PCR primers.

### Data availability

The mass spectrometry proteomics data have been deposited to the ProteomeXchange Consortium via the PRIDE [1] partner repository with the dataset identifier PXD027096 and 10.6019/PXD027096".

The following dataset was generated:

| Author(s) | Year | Dataset title | Dataset URL | Database and Identifier |
|---|---|---|---|---|
| Apte SS | 2022 | Proteolysis of fibrillin-2 microfibrils is essential for normal skeletal development | https://www.ebi.ac.uk/pride/archive/projects/PXD027096 | PRIDE, PXD027096 |

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
