## [Editor Report]

The study proves that the in vivo genetic interaction between ADAMTS6 and fibrillin2 is critical for normal endochondral bone development. In particular, the authors show that global loss of ADAMTS6 causes a severe chondrodysplasia that is significantly worsened by concomitant loss of ADAMTS10 and, conversely, almost fully prevented by haploinsufficiency for fibrillin2, a substrate of ADAMTS6. The paper expands and deepens our current understanding of proteases and their substrates in endochondral bone development.

---

## [Decision Letter]

**Decision letter after peer review:**

Thank you for submitting your article "Proteolysis of fibrillin-2 microfibrils is essential for normal skeletal development" for consideration by *eLife*. Your article has been reviewed by 3 peer reviewers, one of whom is a member of our Board of Reviewing Editors, and the evaluation has been overseen by Kathryn Cheah as the Senior Editor. The following individual involved in review of your submission has agreed to reveal their identity: Karen Lyons (Reviewer #3).

Essential revisions:

Numerous technical issues should be addressed to strengthen the authors' conclusions and their biological relevance.

Please, address reviewers' comments and concerns point-by-point.

*Reviewer #1 (Recommendations for the authors):*

1. The analysis of the skull phenotype, including the cleft palate, is very superficial, incomplete and not properly developed. The authors should consider reporting the correct analysis of the skull in an independent manuscript.

2. Figure 1, supplement 5 is confusing and of poor quality. The authors should provide a systematic histological analysis at different time points and across all the genotypes. It is sufficient to focus exclusively on one skeletal element and systematically analyze it at different time points and in all the genotypes.

3. Along those lines, it would be useful to show whether the appearance of hypertrophy is delayed by loss of ADAMTS6, and this phenotype is worsened by concomitant loss of ADAMTS10.

4. Figure 2: Proper quantification of in situ data should be provided.

5. Figure 3: As above, proper quantification of immunofluorescence data should be provided.

6. Figure 4: Please, provide quantification.

7. Figure 4 , supplement 1: Please, provide quantification.

8. Figure 7: Please, provide quantification of in situ data.

9. It is critical to histologically document whether haploinsufficiency for fibrillin2 prevents the delay of endochondral bone development secondary to loss of ADAMTS6.

*Reviewer #2 (Recommendations for the authors):*

1. Figure 3 and Figure 4. Quantification of Fbn2 abundance (e.g. using primary cells and immunoblot analysis) is desired.

2. Figure 7. Please describe/show the basal phenotype of Fbn2+/- mice. Also, Fbn2 in Fbn2+/- should be quantified.

3. Figure 1A. The transcriptional upregulation of Adamts6 in Adamts10 KO mice. I think that this was previously reported, but is the mechanism known? Is it due to the reduced BMP signaling?

4. The reduced BMP signaling. This was also reported in Adamts10 mutant mice, but has the pathologic significance of the reduced BMP signaling been experimentally tested? Is it just a consequence or does it also contribute to the pathogenesis?

*Reviewer #3 (Recommendations for the authors):*

Figure 1A: There appears to be some upregulation of Adamts6 mRNA in Adamts6 mutants in the heart. Is this significant? Compensatory upregulation?

Figure 1A: The fold increase in Adamts6 mRNA in Adamts10 mutants appears to be modest (lung) or variable (other tissues). Does this increase equate to a difference in protein expression levels? And/or activity?

Figure 1A: The conclusion that Adamts10 mRNA levels are not altered in Adamts6 mutants is based on 3 samples, 2 of which seem to show a reasonable increase in Adamts10 mRNA in Adamts6 mutants. This raises a possibility that there is reciprocal transcriptional regulation in the limb. Please comment on this, given that the phenotypic analysis here was conducted primarily on limb elements.

Figure 1: Please indicate numbers of embryos used in the clear skeletal preparation analysis.

Page 7-Line 174: remove referrals to Figure 1 Suppl. 2 and Figure 1 Suppl 5C-not relevant to the statement about overall lengths of bones.

Page 8 Lines 182-184. The conclusion that ADAMTS6 and ADAMTS10 exhibit cooperative functions is supported by the phenotypes of single and double mutants. However, it is hard to appreciate how the data presented at this point in the manuscript strongly support the conclusion that transcriptional regulation plays a role. The argument that Adamts10-/- mice are indistinguishable from Adamts6+/-;10-/- mice is based only on length measurements, and there are no data correlating phenotypes with Adamts6 mRNA levels in Adamts10-/- and Adamsts6+/-;10-/- mice.

Figure 1 Suppl. 6 and Page 8 line 192: Although the expression of Adamts6 in resting and proliferating chondrocytes is clear at E14.5, it is harder to detect at E13.5 in these cells. Perhaps showing a control image would help? Or using arrows to identify the isolated positive cells at this stage?

Figure 1 Suppl 6 and Page 8 line 195: Arrows/arrowheads should be used to document expression of Adamts6 in tendon/muscle.

Related to this, do the authors mean the articular surface when referring to Adamts10 expression in the joint interzone?

Page 9 Lines 208-209. Although the overall levels of Acan are lower in the columnar zone in the double mutant compared to Adamts6-/- mice, there appears to be more Acan staining in the double mutant compared within the resting zone (RZ). Is this a consistent feature? If so, does it reflect premature differentiation of RZ cells in the double mutant compared to Adamts6-/- mice?

Page 9 Line 211: It looks like *Sox9* expression in the HC is also expanded in all mutants, consistent with delayed ossification.

Figure 2: Were any histological changes noted in the perichondrium, where Adamts6 and Adamts10 are co-localized?

Page 9 Line 216: This is just a suggestion. A statement summarizing the key conclusions from Figure 2 would be helpful. It seems that all mutants exhibited evidence of delayed ossification, but Adamts6-/- mice also exhibited a disorganized growth plate accompanied by reduced matrix expression, which was exacerbated in double mutants.

Figure 3B: It would be best to highlight the posterior femur perichondrium in Adamts6-/- mutants since that is the region used for all other panels in Figure 3A and B. Also, an arrow to inform readers where the perichondrium is will help to discriminate this tissue from the adjacent connective tissue, where staining also appears to be elevated in mutants.

Page 10 Lines 226-7: The statement is correct, but it seems that there is no expression of fibrillin-1 in perichondrium. The staining appears to be in adjacent connective tissue.

Figure 5 Suppl. 1: This is not a major point, but it might be helpful to the reader to show the schematics of both the Fbn1- and Fb22-Nt and -Ct fragments to help interpret Figure 5 Suppl. 1C.

Page 12 Line 297: Check journal policy regarding "data not shown."

Page 14 Lines 340-342: It is concluded that ADAMTS6 has a selective impact on fibrillin-2. However, Figure 3 shows little to no Fibrillin-1 staining within the growth plate or perichondrium. Therefore, the selective impact could be due to the much greater abundance of fibrillin-2.

Page 14 Lines 346-7 and Figure 7 supple 7A. RZ is shown for WT and Adamts6-/- sections, but the CZ is shown for Adamts6-/-;Fn2+/- mice. Show the same zone for all genotypes in the magnifications.

Page 10 Line 4310-436: I am not convinced that the data document that the observed upregulation of Adamts6 mRNA in the Adamts10 mutant is the reason the latter mutants have a less severe phenotype. Perhaps a protease other than Adamts6 compensates for loss of Adamts10.

---

## [Author Response]

Reviewer #1 (Recommendations for the authors):1. The analysis of the skull phenotype, including the cleft palate, is very superficial, incomplete and not properly developed. The authors should consider reporting the correct analysis of the skull in an independent manuscript.

The phenotype in the skull (Figure 1 —figure supplement 10) illustrates along with the defective ribs, sternum and vertebrae that development of the axial skeleton, and not only the appendicular skeleton, is perturbed in the absence of ADAMTS6, emphasizing a global impact on skeletal development, whose mechanisms are specifically investigated in the appendicular skeleton. The incidence of cleft palate in *Adamts6*^-/-^ embryos (Figure 8 —figure supplement 2B) is statistically significant and is important to retain, because like long bone defects in *Adamts6*^-/-^ embryos, it is consistently reversed to normalcy in *Adamts6*^-/-^;*Fbn2*^+/-^ embryos. Along with the restored normal architecture of the axial and appendicular skeleton, the reversal of cleft palate provides key data supporting reduced fibrillin-2 proteolysis as a primary mechanism of *Adamts6*. We have therefore kept this data as part of the manuscript as a supplemental figure.

2. Figure 1, supplement 5 is confusing and of poor quality. The authors should provide a systematic histological analysis at different time points and across all the genotypes. It is sufficient to focus exclusively on one skeletal element and systematically analyze it at different time points and in all the genotypes.

We thank the reviewer for this comment. We have replaced Figure 1, supplement 5 with new figures (Figure 1 —figure supplement 6, 7, 8, and 9). We also include additional new analysis at E14.5 time point, to demonstrate the role of ADAMTS6 at an earlier landmark in skeletal development when primary ossific nuclei first appear (Figure 3 —figure supplement 2).

3. Along those lines, it would be useful to show whether the appearance of hypertrophy is delayed by loss of ADAMTS6, and this phenotype is worsened by concomitant loss of ADAMTS10.

We thank the reviewer for pointing this out. In early bone development leading up to appearance of primary ossific centers, it does appear that hypertrophy is generally delayed, although chondrocyte hypertrophy is not evident in the developing *Adamts6*^-/-^ fibula at E16.5, whereas it is evident at 14.5 in the wild type (Figure 1 —figure supplement 8). Once these centers are formed, the hypertrophic zones were comparable as indicated by collagen X staining (Figure 3B).

4. Figure 2: Proper quantification of in situ data should be provided.8. Figure 7: Please, provide quantification of in situ data.

The in situ analysis of *Adamts6* and *Adamts10* in wild-type tissues is provided to show the spatial distribution of mRNA of these two genes. We have expanded this dataset to include additional time points to emphasize that both genes are expressed in the perichondrium and peripheral cartilage, developing skeletal muscle, tendon and connective tissue (Figure 2). While we have quantified all immunofluorescence in the manuscript, we have not quantified the in situ data, since it was not intended to be comparative between genotypes.

5. Figure 3: As above, proper quantification of immunofluorescence data should be provided.6. Figure 4: Please, provide quantification.7. Figure 4 , supplement 1: Please, provide quantification.

As recommended, all immunofluorescence staining has now been quantified and data displayed in bar graphs in Figure Supplements corresponding to figures where immunofluorescence is shown.

9. It is critical to histologically document whether haploinsufficiency for fibrillin2 prevents the delay of endochondral bone development secondary to loss of ADAMTS6.

We agree that this is an essential point. As Figure 8B shows, there is comparable endochondral ossification in the *Adamts6*^-/-^;*Fbn2*^+/-^ hindlimbs as the wild type, and distinctly more than in *Adamts6*^-/-^ mutant limbs in all bones, specifically in the fibula, for example, where it was most drastically reduced. In addition, histologically, proteoglycan content and *Sox9* staining were significantly restored toward normal based on immunofluorescence staining of sections (Figure 8D).

Reviewer #2 (Recommendations for the authors):1. Figure 3 and Figure 4. Quantification of Fbn2 abundance (e.g. using primary cells and immunoblot analysis) is desired.

We have quantified all primary cell and immunoblot data (Figure 4 —figure supplement 1A, Figure 5 and Figure 5 —figure supplement 1).

2. Figure 7. Please describe/show the basal phenotype of Fbn2+/- mice. Also, Fbn2 in Fbn2+/- should be quantified.

In response, this has been addressed experimentally (Figure 8 —figure supplement 3 and 4).

3. Figure 1A. The transcriptional upregulation of Adamts6 in Adamts10 KO mice. I think that this was previously reported, but is the mechanism known? Is it due to the reduced BMP signaling?

The transcriptional upregulation of *Adamts6* (Figure 1A)was not previously shown in *Adamts10*^-/-^ mice or in any other in vivo system; prior work showed a similar upregulation in vitro in an ocular cell line (ARPE-19) upon *Adamts6* knockdown (Cain et al., *Sci Rep.* 2016, PMID 27779234), but not in vivo. Reciprocal transcriptional upregulation of *Adamts10* in *Adamts6*^-/-^ mice is also shown in our study, which is a further novel finding. BMPs are not known to have a role in this process and we did not specifically investigate its basis.

4. The reduced BMP signaling. This was also reported in Adamts10 mutant mice, but has the pathologic significance of the reduced BMP signaling been experimentally tested? Is it just a consequence or does it also contribute to the pathogenesis?

The previous report showed that BMP signaling was reduced in fibroblasts isolated from an *Adamts10* genetic mutant disease model, but not in mouse tissues/skeleton (Mularczyk et al., Human Mol Genet, 2018, PMID 30060141). However, the mechanism of the reduction was not explored at all nor was it ever previously analyzed in *Adamts6*^-/-^ mutants. We show here that the reduction of BMP signaling (as shown by reduced pSMAD1/5/8 signaling both in tissue sections and via western blot analysis) in *Adamts6*^-/-^ cartilage is due to the increase of fibrillin-2 and that, in turn, results in sequestration of GDF5 (Figure 9, Figure 10, Figure 10 —figure supplement 1 and Figure 11). Upon introduction of *Fbn2* haploinsufficiency, the reduced BMP signaling in *Adamts6*^-/-^ mutants was clearly mitigated (Figure 9A-D).

Reviewer #3 (Recommendations for the authors):Figure 1A: There appears to be some upregulation of Adamts6 mRNA in Adamts6 mutants in the heart. Is this significant? Compensatory upregulation?Figure 1A: The fold increase in Adamts6 mRNA in Adamts10 mutants appears to be modest (lung) or variable (other tissues). Does this increase equate to a difference in protein expression levels? And/or activity?Figure 1A: The conclusion that Adamts10 mRNA levels are not altered in Adamts6 mutants is based on 3 samples, 2 of which seem to show a reasonable increase in Adamts10 mRNA in Adamts6 mutants. This raises a possibility that there is reciprocal transcriptional regulation in the limb. Please comment on this, given that the phenotypic analysis here was conducted primarily on limb elements.

Thank you for your comments regarding the qPCR analysis of *Adamts6* and *Adamts10* in the mutant embryo tissues. We originally had n=3 for each genotype and tissue. We have subsequently doubled this to n=6 to address the reviewer comment. In this revised dataset, there is decreased *Adamts6* mRNA in *Adamts6*^-/-^ limbs and lungs, but not the heart. The fold increase in *Adamts6* mRNA in *Adamts10*^-/-^ embryos is 2-, 4- and 2.5-fold in the limb, heart and lung, respectively. This suggests that the ADAMTS6 upregulation could provide a potential albeit partial compensatory mechanism in *Adamts10* mutant mice. We also observed compensatory upregulation of *Adamts10* mRNA in *Adamts6*^-/-^ mice. We conclude, as the reviewer suggested, that there is reciprocal transcriptional regulation in the limb. In addition, we quantified *Adamts17* and *Adamts19* mRNAs since mutations of ADAMTS17, like those in ADAMTS10, can cause Weill-Marchesani syndrome (Morales et al., PMID19836009). ADAMTS19 was included as the closest homolog of ADAMTS17. There was no alteration of either *Adamts17* or *Adamts19* mRNA levels in the mutant tissues, suggesting specificity of the ADAMTS6 and ADAMTS10 response in relation to each other (updated Figure 1A).

Figure 1: Please indicate numbers of embryos used in the clear skeletal preparation analysis.

The n of each genotype is shown in Figure 1 —figure supplement 4.

Page 7-Line 174: remove referrals to Figure 1 Suppl. 2 and Figure 1 Suppl 5C-not relevant to the statement about overall lengths of bones.

Thank you for this comment. This has been revised as suggested.

Page 8 Lines 182-184. The conclusion that ADAMTS6 and ADAMTS10 exhibit cooperative functions is supported by the phenotypes of single and double mutants. However, it is hard to appreciate how the data presented at this point in the manuscript strongly support the conclusion that transcriptional regulation plays a role. The argument that Adamts10-/- mice are indistinguishable from Adamts6+/-;10-/- mice is based only on length measurements, and there are no data correlating phenotypes with Adamts6 mRNA levels in Adamts10-/- and Adamsts6+/-;10-/- mice.

We suggest that ADAMTS6 and ADAMTS10 exhibit cooperative functions due to the more drastic phenotype in the double-knockout embryo as compared to either single knockout embryos (Figure 1B). The observed reciprocal transcriptional regulation of these homologous proteases (Figure 1A) could have a role in masking the individual phenotype, but we agree with the reviewer that this remains open to interpretation. We have softened this point accordingly to state that although transcriptional adaptation is observed and could mask the phenotypes of individual gene mutants, its precise impact remains unclear.

Figure 1 Suppl. 6 and Page 8 line 192: Although the expression of Adamts6 in resting and proliferating chondrocytes is clear at E14.5, it is harder to detect at E13.5 in these cells. Perhaps showing a control image would help? Or using arrows to identify the isolated positive cells at this stage?

Thank you for the suggestion. We now show expression at E13.5 and E14.5 at higher magnification for both *Adamts6* and *Adamts10* throughout the musculoskeletal system (Figure 2). The recommended control for RNAscope (*DapB* probe) did not give a signal, but is not illustrated.

Figure 1 Suppl 6 and Page 8 line 195: Arrows/arrowheads should be used to document expression of Adamts6 in tendon/muscle.Related to this, do the authors mean the articular surface when referring to Adamts10 expression in the joint interzone?

As suggested, we have added specific annotation of tendons and muscle in in situ panels (Figure 2). *Adamts10* is broadly expressed throughout the musculoskeletal elements, which include the joint interzone as well as the surrounding tissues. At this stage, a synovial cavity is not apparent and we designated the inter-cartilage mesenchyme as constituting the joint interzone.

Page 9 Lines 208-209. Although the overall levels of Acan are lower in the columnar zone in the double mutant compared to Adamts6-/- mice, there appears to be more Acan staining in the double mutant compared within the resting zone (RZ). Is this a consistent feature? If so, does it reflect premature differentiation of RZ cells in the double mutant compared to Adamts6-/- mice?

There is an overall reduction of Acan both in *the Adamts6*^-/-^ and the *Adamts6*^-/-^;*Adamts10*^-/-^ growth plates (Figure 3 —figure supplement 3 shows no statistical difference between the two groups).

Page 9 Line 211: It looks like Sox9 expression in the HC is also expanded in all mutants, consistent with delayed ossification.

We thank the reviewer for noting this. *Sox9* is not only expressed in the early chondrocytes of the resting zone, but also in hypertrophic chondrocytes as pointed out, but we quantified it in the reserve and proliferative zones where the matrix was greatest.

Figure 2: Were any histological changes noted in the perichondrium, where Adamts6 and Adamts10 are co-localized?

In response to this comment, we include data showing altered perichondrium ECM (more collagen staining) using RGB trichrome (Figure 3 —figure supplement 1). In addition, as shown with fibrillin-2 and MAGP-1 staining, there is an increase of both molecules in perichondrial ECM (Figure 4, Figure 10).

Page 9 Line 216: This is just a suggestion. A statement summarizing the key conclusions from Figure 2 would be helpful. It seems that all mutants exhibited evidence of delayed ossification, but Adamts6-/- mice also exhibited a disorganized growth plate accompanied by reduced matrix expression, which was exacerbated in double mutants.

We agree and have added a concluding sentence in this section (top of pg 10), stating “These findings therefore indicate a substantial disruption of the *Adamts6*-mutant chondrocyte phenotype, which was exacerbated in the *Adamts6*^-/-^;*Adamts10*^-/-^ mutants”.

Figure 3B: It would be best to highlight the posterior femur perichondrium in Adamts6-/- mutants since that is the region used for all other panels in Figure 3A and B. Also, an arrow to inform readers where the perichondrium is will help to discriminate this tissue from the adjacent connective tissue, where staining also appears to be elevated in mutants.

We have updated the figure (now Figure 4) to indicate the posterior perichondrium.

Page 10 Lines 226-7: The statement is correct, but it seems that there is no expression of fibrillin-1 in perichondrium. The staining appears to be in adjacent connective tissue.

We thank the reviewer for pointing this out. We agree, the staining is primarily in the adjacent connective tissue, which is unchanged in the mutants as compared to the control (Figure 4C).

Figure 5 Suppl. 1: This is not a major point, but it might be helpful to the reader to show the schematics of both the Fbn1- and Fb22-Nt and -Ct fragments to help interpret Figure 5 Suppl. 1C.

We have updated the figures accordingly (Figure 6A and Figure 6 —figure supplement 1A).

Page 12 Line 297: Check journal policy regarding "data not shown."

As per journal policy, we have deleted all instances of “data not shown” from the manuscript.

Page 14 Lines 340-342: It is concluded that ADAMTS6 has a selective impact on fibrillin-2. However, Figure 3 shows little to no Fibrillin-1 staining within the growth plate or perichondrium. Therefore, the selective impact could be due to the much greater abundance of fibrillin-2.

We thank the reviewer for this observation and agree. The impact on fibrillin-2 is greater, consistent with this isoform being more abundant in the embryonic period (Figure 4 and quantified in Figure 4 —figure supplement 1A) and our conclusion that a major function of ADAMTS6 is to control/reduce the fibrillin-2 microfibril burden as development progresses toward postnatal life.

Page 14 Lines 346-7 and Figure 7 supple 7A. RZ is shown for WT and Adamts6-/- sections, but the CZ is shown for Adamts6-/-;Fn2+/- mice. Show the same zone for all genotypes in the magnifications.

Thank you for pointing this out. The figure (now Figure 8C) is revised accordingly.

Page 10 Line 4310-436: I am not convinced that the data document that the observed upregulation of Adamts6 mRNA in the Adamts10 mutant is the reason the latter mutants have a less severe phenotype. Perhaps a protease other than Adamts6 compensates for loss of Adamts10.

We agree that the viewer’s interpretation is justified and have revised the text to provide a more nuanced interpretation. ADAMTS10 is weakly activated by furin and promotes fibrillin-1 microfibril assembly by cultured fibroblasts, as demonstrated by us and others (Kutz et al., J Biol Chem, 2011, PMID 21402694; Mularczyk et al., Human Mol Genet, 2018, PMID 30060141). It cleaves fibrillin-2, but only weakly cleaves fibrillin-1 and only when activated by optimizing the furin cleavage site (Kutz et al., J Biol Chem, 2011, PMID 21402694;Wang et al., Matrix Biology 2019, PMID 30201140). Hence, ADAMTS10 differs from ADAMTS6 in these important characteristics, and a relevant experimental observation highlighted in the present work is that both fibrillin-1 and fibrillin-2 microfibrils are absent in the presence of ADAMTS6 (Figure 5). In response to the reviewers second comment, we determined mRNA levels of homologous proteases *Adamts17* and *Adamts19* and observed no change in expression of either in the mutants (Figure 1A).